# Inferring multimodal latent topics from electronic health records

Yue Li [1✉], Pratheeksha Nair[1], Xing Han Lu[1], Zhi Wen[1], Yuening Wang[1], Amir Ardalan Kalantari Dehaghi[1], Yan Miao[1], Weiqi Liu[1], Tamas Ordog [2], Joanna M. Biernacka [3,4], Euijung Ryu[3], Janet E. Olson [3], Mark A. Frye[4], Aihua Liu[5], Liming Guo[5], Ariane Marelli[5], Yuri Ahuja[6], Jose Davila-Velderrain[6] & Manolis Kellis [6,7✉]

Electronic health records (EHR) are rich heterogeneous collections of patient health information, whose broad adoption provides clinicians and researchers unprecedented opportunities for health informatics, disease-risk prediction, actionable clinical recommendations, and precision medicine. However, EHRs present several modeling challenges, including highly sparse data matrices, noisy irregular clinical notes, arbitrary biases in billing code assignment, diagnosis-driven lab tests, and heterogeneous data types. To address these challenges, we present MixEHR, a multi-view Bayesian topic model. We demonstrate MixEHR on MIMIC-III, Mayo Clinic Bipolar Disorder, and Quebec Congenital Heart Disease EHR datasets. Qualitatively, MixEHR disease topics reveal meaningful combinations of clinical features across heterogeneous data types. Quantitatively, we observe superior prediction accuracy of diagnostic codes and lab test imputations compared to the state-of-art methods. We leverage the inferred patient topic mixtures to classify target diseases and predict mortality of patients in critical conditions. In all comparison, MixEHR confers competitive performance and reveals meaningful disease-related topics.

[1] School of Computer Science and McGill Centre for Bioinformatics, McGill University, Montreal, Quebec H3A0E9, Canada. [2] Department of Physiology and Biomedical Engineering and Division of Gastroenterology and Hepatology, Department of Medicine, and Center for Individualized Medicine, Mayo Clinic, Rochester, MN, USA. [3] Department of Health Sciences Research, Mayo Clinic, Rochester, MN, USA. [4] Department of Psychiatry and Psychology, Mayo Clinic, Rochester, MN, USA. [5] McGill Adult Unit for Congenital Heart Disease Excellence (MAUDE Unit), Montreal QC H4A 3J1 Quebec, Canada. [6] Computer Science and Artificial Intelligence Lab, Massachusetts Institute of Technology, 32 Vassar St, Cambridge, MA 02139, USA. [7] The Broad Institute of Harvard and MIT, 415 Main Street, Cambridge, MA 02142, USA. ✉email: yueli@cs.mcgill.ca; manoli@mit.edu

The broad adoption of electronic health record (EHR) systems has created unprecedented resources and opportunities for conducting health informatics research. Hospitals routinely generate EHR data for millions of patients, which are increasingly being standardized by using systematic codes, such as Logical Observation Identifiers Names and Codes (LOINC) for lab tests, RxNorm for prescriptions; and Systematized Nomenclature of Medicine (SNOMED), Diagnosis-Related Groups (DRG), and International Classification of Diseases (ICD-9) for diagnoses (albeit for billing purposes). In the USA, for example, the number of nonfederal acute care hospitals with basic digital systems increased from 9.4% to 96% over the 7-year period between 2008 and 2015[1–3]. Furthermore, the amount of comprehensive EHR data recording multiple data types, including clinical notes, increased from only 1.6% in 2008 to 40% in 2015[3]. With the aid of effective computational methods, these EHR data promise to define an encyclopedia of diseases, disorders, injuries, and other related health conditions, uncovering a modular phenotypic network.

Distilling meaningful concepts from the raw EHR data presents several challenges, and it is often unfeasible to directly model the joint distribution over the entire EHR feature space. On the other hand, it is possible to formulate a latent topic model over discrete data, in analogy to automatic text categorization[4], considering each patient as a document and each disease meta-phenotype as a topic (grouping recurrent combinations of individual phenotypes). Our task then is to learn a set of meaningful disease topics, and the probabilistic mixture memberships of patients for each disease topic, representing the combination of meta-phenotypes inferred for each patient.

In this paper, we formalize this analogy by introducing a latent topic model, MixEHR, designed to meet the challenges intrinsic to heterogeneous EHR data. The main objective of our approach is twofold: (1) distill meaningful disease topics from otherwise highly sparse, biased, and heterogeneous EHR data; and (2) provide clinical recommendations by predicting undiagnosed patient phenotypes based on their disease mixture membership. Importantly, we aim for our model to be interpretable in that it makes not only accurate predictions but also intuitive biological sense. MixEHR can simultaneously model an arbitrary number of EHR categories with separate discrete distributions. For efficient Bayesian learning, we developed a variational inference algorithm that scales to large-scale EHR data.

MixEHR builds on the concepts of collaborative filtering[5–9] and latent topic modeling[4,10–14]. In particular, our method is related to the widely popular text-mining method Latent Dirichlet Allocation (LDA)[4]. However, LDA does not account for missing data and NMAR mechanism[8,15]. Our method is also related to several EHR phenotyping studies focusing on matrix factorization and model interpretability[16–23]. Recently developed deep-learning methods primarily focus on prediction performance of target clinical outcomes[24–31]. Apart from these methods, Graph-based Attention Model (GRAM)[32] uses the existing knowledge graph to embed ICD code and make prediction of diagnostic codes at the next admission using the graph embedding. Deep patient[29] uses a stacked stochastic denoising autoencoder to model the EHR code by its latent embedding, which are used as features in a classifier for predicting a target disease. More detailed review are described in Supplementary Discussion.

We apply MixEHR to three real-world EHR datasets: (1) Medical Information Mart for Intensive Care (MIMIC)-III[1]: a public dataset to date containing over 50,000 ICU admissions; (2) Mayo Clinic EHR dataset containing 187 patients, including with 93 bipolar disorder and 94 controls; (3) The Régie de l'assurance maladie du Québec Congenital Heart Disease Dataset (Quebec CHD Database) on over 80,000 patients being followed for congenital heart disease (CHD) for over 28 years (1983–2010) at the McGill University Health Centre (MUHC) in Montreal. In these applications, we find that the clinical features emerging across EHR categories under common disease topics are biologically meaningful, revealing insights into disease co-morbidities. The inferred topics reveal sub-categories of bipolar disorder, despite the small sample size. The inferred patient topic mixtures can be used to effectively predict diagnostic code in patient's future hospital visits from outpatient data, impute missing lab results, and predict future next-admission patient mortality risk.

## Results

**MixEHR probabilistic model**. We take a probabilistic joint matrix factorization approach, by projecting each patient's high-dimensional and heterogeneous clinical record onto a low-dimension probabilistic meta-phenotype signature, which reflects the patient's mixed memberships across diverse latent disease topics. The key innovation that enables us to handle highly heterogeneous data types is that we carry out this factorization at two levels. At the lower level, we use data-type-specific topic models, learning a set of basis matrices for each data type. Using the MIMIC-III data as an particular example, we learned seven basis matrices corresponding to clinical notes, ICD-9 billing codes, prescriptions, DRG billing codes, CPT procedural codes, lab tests, and lab results from the MIMIC-III dataset. We link these seven basis matrices at the higher level using a common loading matrix that connects the multiple data types for each patient (Fig. 1a).

To model the lab data, we assign common latent topic to both the lab test and lab test result such that they are conditionally independent given their topic assignments (Fig. 1b). We then updates the lab topic distributions using both the observed lab test results and the missing/imputed lab test results weighted by their topic assignment probabilities for each patient. This differs from other latent topic models, which assume that data are either completely observed or missing at random. To learn the model, we use a variational Bayes to update each model parameters in a coordinate ascent (Fig. 1c). The inference algorithms are detailed in "Methods" and "Supplementary Methods". After learning the model, we can obtain three pieces of clinically relevant information: (1) to impute missing EHR code for each patient, we take average over the disease topics; (2) to infer patients' meta-phenotypes as their mixed disease memberships, we average over the phenotype dimension; and (3) to infer latent disease topic distribution (i.e., distributions over all clinical variables), we average over the patient dimension.

**Multimodal disease topics from MIMIC-III**. We applied MixEHR to MIMIC-III data[1], which contain ~39,000 patients each with a single admission and ~7500 patients each with multiple admissions (Supplementary Table 1). We used MixEHR to jointly model six data categories, including unstructured text in clinical notes, ICD-9, DRG, current procedural terminology (CPT), prescriptions, and lab tests, together comprising of ~53,000 clinical features and ~16 million total clinical observations ("Methods"). The frequency of observed EHR code over all admissions is low (Supplementary Fig. 27). Majority of the variables including the lab tests were observed in less than 1% of the 58,903 admissions (i.e., greater than 99% missing rates for most variables). Therefore, the data are extremely sparse underscoring the importance of integrating multimodal data information to aid the imputation.

To choose the optimal number of topics (K), we performed fivefold cross-validation to evaluate models with different K by the averaged predictive likelihood on the held-out patients (Supplementary Fig. 1a). We set K = 75 for subsequent analyses as it gave the highest predictive likelihood. We then examined the

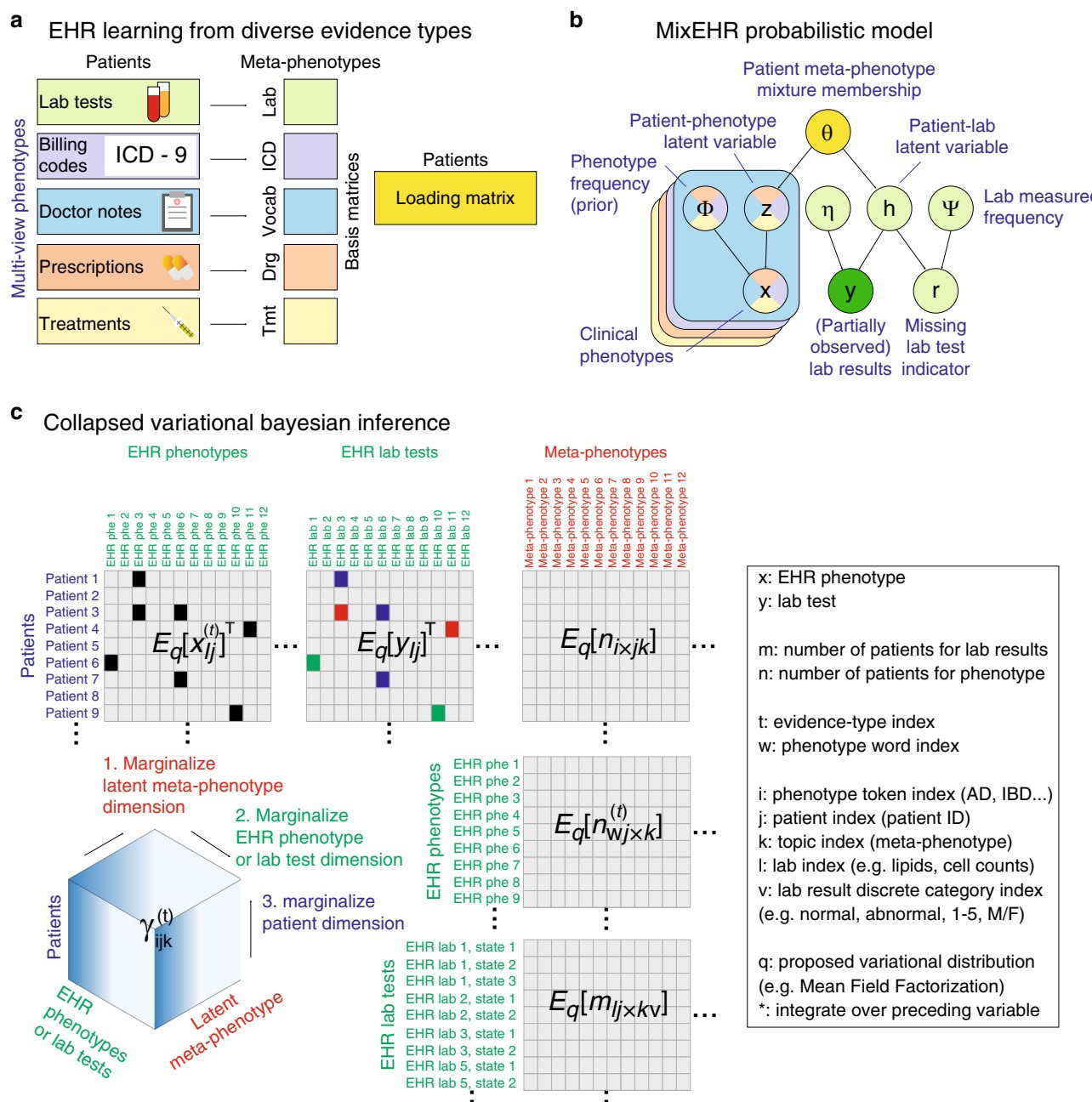

**Fig. 1 MixEHR model overview. a** Multi-view matrix factorization of multiple data matrices corresponding to different EHR data types, including lab tests, billing code, doctor notes, etc. **b** Proposed Bayesian model for modeling non-missing-at-random (NMAR) lab tests and other multimodal data. In order to achieve tractable inference, we assign a latent topic $h_{lj}$ to the lab results $y_{lj}$ and missing indicator ($r_{lj}$) such that they become conditionally independent. **c** Collapsed variational Bayesian inference of the MixEHR model. The inference and learning can be visualized as marginalizing a three-dimensional tensor that represents the expectations of the latent variables.

clinical relevance of our learned disease topics by qualitatively assessing the coherence of the 5 most probable EHR codes for each topic. Taking a column with common index $k$ from the topic distribution matrices for each of the six EHR data types (i.e., notes, ICD-9, CPT, DRG, lab tests, and prescription) gives us a single disease topic distribution over clinical terms across those data types. We annotated all of the 75 topics based on the top EHR codes and found that majority of the inferred topics are specific to distinct diseases (Supplementary Table 2).

For the purpose of demonstration, we paid special attention to the disease topics exhibiting the highest likelihoods over the broad set of ICD-9 codes with prefixes {250, 296, 331, 042, 205, 415, 571}, representing diabetes, psychoses, neurodegeneration, HIV, myeloid leukemia, acute pulmonary heart disease, and chronic liver disease and cirrhosis, respectively. The top terms are displayed in alternative word cloud representations in Supplementary Fig. 3. We observed salient intra-topic consistency and inter-topic contrast with respect to each topic's highest-scoring EHR features across diverse EHR data categories (colored bars on the right; Fig. 2a). Notably, if we were to use LDA and thereby model observations across all data categories as draws from the same multinomial distribution, the learned disease topics would have been dominated by clinical notes, given that these contain far more features (i.e., words) than the other categories. We

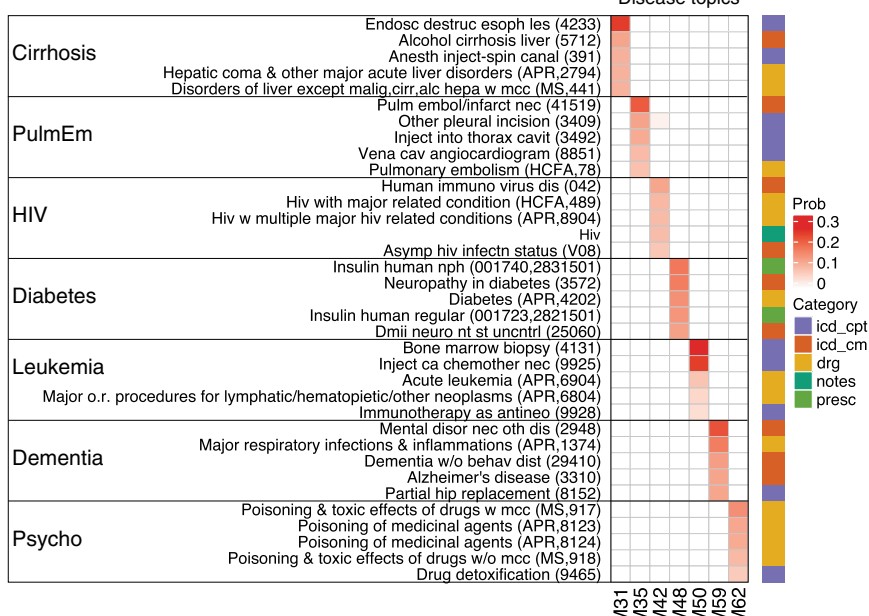

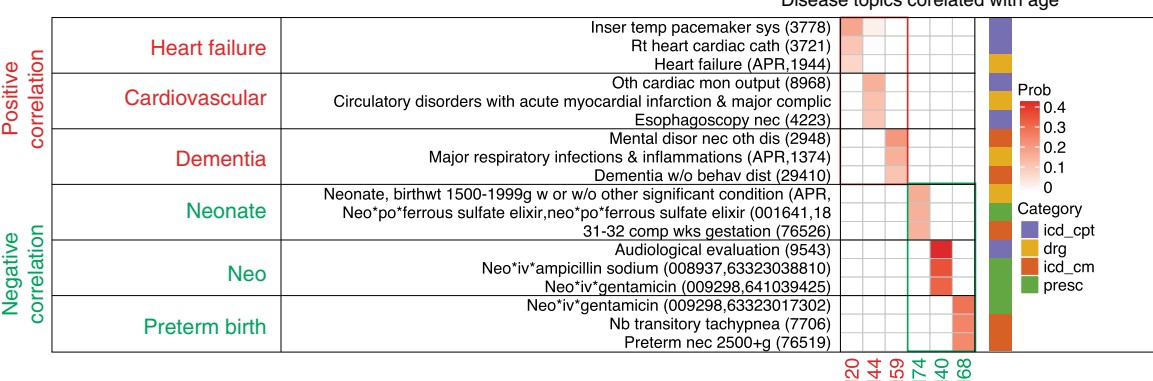

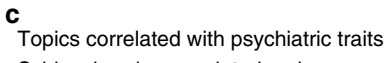

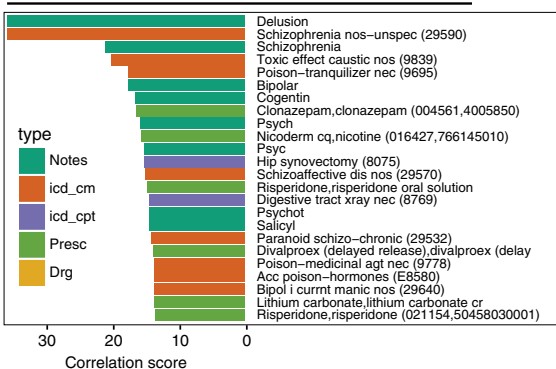

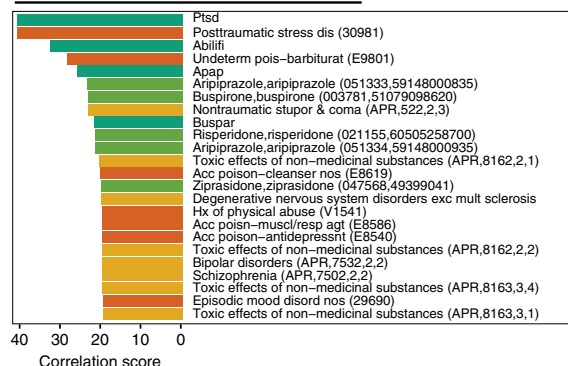

**Fig. 2 Disease topics from MIMIC-III dataset. a** Top five EHR codes for select disease topics from the 75-topic MixEHR model. The rows are the top features, and columns are the topics. The color intensities indicate the inferred probabilities of these features belonging to each topic. **b** Age-correlated disease topics. Top three EHR codes were displayed for the top three most and least age-correlated topics. **c** Top correlated EHR codes with schizophrenia and PTSD.

illustrate this aspect in the application of mortality prediction below.

Moreover, we observed interesting connections between abnormal lab results and specific diseases (Supplementary Fig. 4). For instance, topic M31 is enriched for alcoholic cirrhosis billing codes and ascites-based lab tests typical for cirrhosis patients; in M42, the code for HIV infection is grouped with abnormal counts of immune cells, including CD4 lymphocytes; diabetes codes tend to rank highly alongside high blood acetone in M48, and topic M62 is enriched for opioid abuse along with valproic acid and lithium prescriptions—common treatments for bipolar disorder. Interestingly, topic M59—associated with neurodegenerative diseases such as Alzheimer's disease (AD)—is strongly associated with vitamin B12, which was recently shown to exhibit differential pattern in AD patients[33]. We also correlated the 75-topic distributions with patients' age. The results clearly show that the top positively age-correlated topics are related to heart failure (M20), cardiovascular (M44), and dementia (M59). In contrast, the top three negatively age-correlated topics are all related with newborn (Fig. 2b; Supplementary Fig. 5).

**Disease comorbidity network in MIMIC-III data**. We reasoned that the similarity of the annotated disease (ICD-9 codes) distributions over topics might help uncover nontrivial associations between diseases. In order to infer a disease comorbidity network, we used the 75 learned disease topics and calculated the pairwise correlations between ICD-9 diagnostic codes. We included 424 ICD-9 codes observed in at least 100 patients, and not in the external injuries (ICD-9 code starting with V) or supplementary classification (ICD-9 code starting with E) sections. For comparison, we calculated the correlations between the same 424 ICD-9 codes using their sparse distributions over patients in the raw data. Despite focusing on the ICD-9 codes with at least 100 patients, on average each code is observed in only 0.8% of patients (min: 0.3% and max 2%). Consequently, due to the high sparsity of diseases among patients, we observed very weak correlations based on the observed data ranging between −0.02 and 0.06 (Supplementary Fig. 6a). However, when we correlated the ICD-9 codes using our inferred topic embeddings, we observed much stronger correlations. More importantly, diseases of similar physiology form modules consistent with clinical intuition (Supplementary Fig. 6b). Thus, the projection of multimodal data to the inferred topic embeddings space enables the discovery of strong disease association, not directly measurable from the raw EHR data.

Using these topic embeddings, we can also obtain a comorbidity network centering on a specific disease of interest by correlating the disease's diagnostic code with all other diagnostic codes. In particular, we obtained such a comorbidity network for schizophrenia, post traumatic stress disorder (PTSD), Alzheimer's disease, and bipolar disorder (Fig. 2c; Supplementary Figs. 7–14). The size of the terms are proportional to their Pearson correlation with the disease of interest. To control the false discovery rate, we randomly shuffled the topic probabilities for the target phenotype to calculate the background correlation. Only phenotypes with permutation $p$-value $< 0.01$ are displayed. We observed meaningful related phenotypes in all of the four examples.

**Patient risk prioritization using MIMIC-III data**. Besides gaining insights from the disease topics (i.e., basis matrices), we can also exploit the disease topic mixture memberships along the patients dimension. As an example, we took the 50 patients with the highest proportions for topics M31, M35, and M50, which as we noted above are associated with alcoholic cirrhosis,

pulmonary embolism, and leukemia, respectively (Fig. 3a). We observed a clear enrichment for the expected ICD-9 codes—571, 415, and 205, corresponding to the diagnoses of chronic liver disease and cirrhosis, acute pulmonary heart disease, and myeloid leukemia, respectively—among these patients (Fig. 3b). However, not all of the patients were classified by their expected diagnostic codes.

We then hypothesized that these unclassified patients perhaps exhibit other markers of the relevant diseases. To further investigate these high-risk patients, we examined the 10 highest-scoring EHR codes under these topics (Fig. 3c). We highlighted some evidence as why the patients with the absence of ICD-9 code were prioritized as high risk by our model. Most patients under the leukemia topic M31, including those missing ICD-9 code 205, received bone marrow biopsies (CPT code 4131) and infusion of a cancer chemotherapeutic substance (CPT code 9925)—clear indicators of leukemia diagnosis and treatment. Likewise, several patients under the pulmonary embolism topic (M35) that are missing ICD-9 code 415 nevertheless have the DRG billing code for pulmonary embolism, a prescription for enoxaparin sodium (which is used to treat pulmonary embolism), or the related ICD-9 code 453 for other venous embolism and thrombosis. Moreover, all patients under the cirrhosis topic M50 have the key word cirrhosis mentioned in their clinical notes. Notably, although the procedural code for esophageal variceal banding is not particularly prevalent, MixEHR predicts reasonably high probability of undergoing this procedure among cirrhosis-topic patients—a clinically reasonable prediction. This suggests that our model is not only intuitively interpretable but also potentially useful for suggesting patient's future medical interventions.

**Classification of Mayo Clinic bipolar disorder patients**. To further demonstrate the utility of our approach in discovering meaningful multimodal topics, we applied MixEHR to a separate dataset containing 187 patients, including 93 bipolar disorder cases and 94 age- and sex-matched controls from Mayo Clinic. Despite the small sample size, the patients were deeply phenotyped: there are in total 7731 heterogeneous EHR features across five different data categories, including ICD-9 codes, procedure codes, patient provided information (PPI), lab tests, and prescription codes. In total, there are 108,390 observations.

We evaluated our model by fivefold cross-validation. Here, we trained MixEHR or the unsupervised baseline models and a logistic regression (LR) classifier that uses the patient topic mixture derived from MixEHR or embeddings derived from other methods to predict the bipolar disorder label (Fig. 4a; "Methods"). We observed superior performance of MixEHR+LR compared with LDA+LR and RBM+LR in both the area under the ROC and the area under the precision–recall curves (Fig. 4b). We quantified the predictive information of the 20 topics based on the linear coefficients of the LR classifier. We observed that topics M19 and M7 have the highest positive coefficients for bipolar disorder label (Fig. 4c). To confirm the finding, we visualized the 187 patient topic mixtures side-by-side with their BD diagnosis ICD-9 296 code (Fig. 4d). Indeed, we find that M19 and M7 are highly correlated with the BD diagnosis label, and M19 and M7 may represent two distinct subgroups of BD patients. To check whether these two topics were driven by demographic information, we compared the topic mixture membership with sex and age for each patient. We did not observe significant difference between sexes for M19 mixture probabilities or M7 mixture probabilities (Supplementary Fig. 28). There is also no significant correlation between age and M19 or M7 topic mixture (Supplementary Fig. 31).

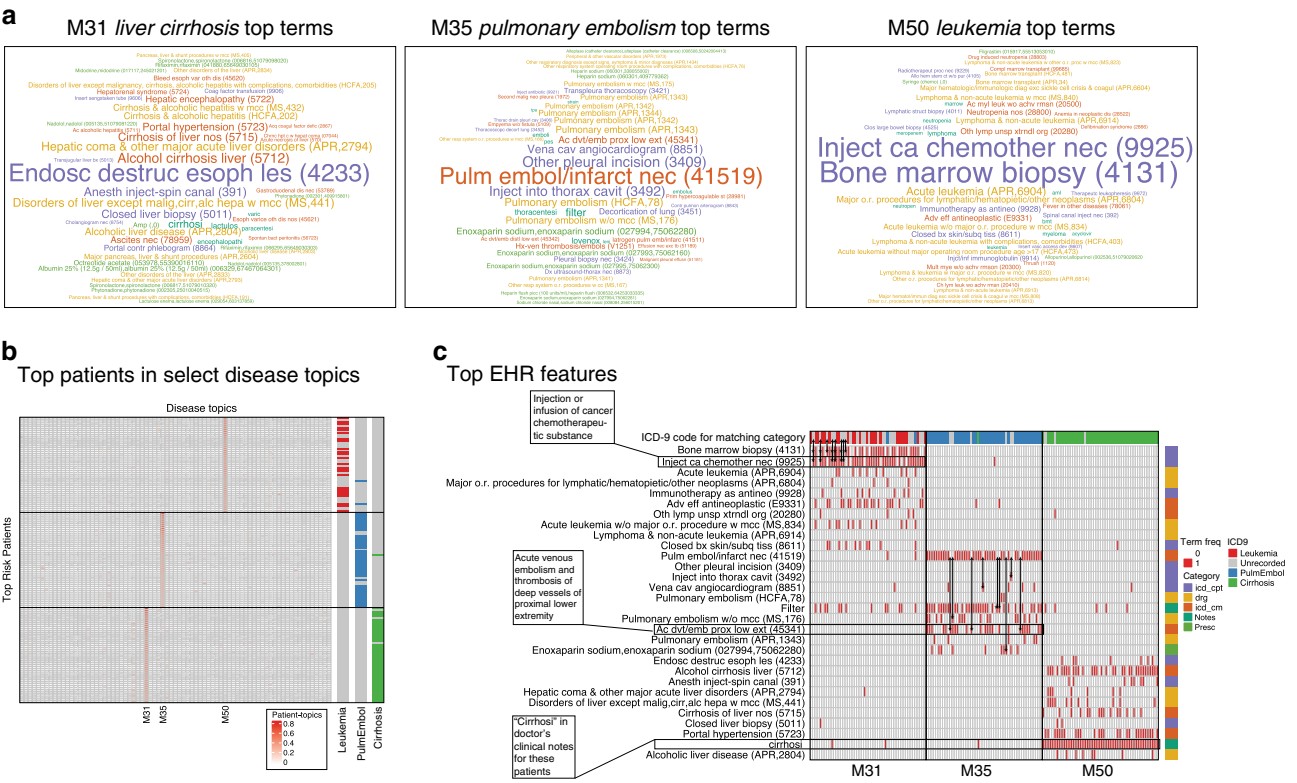

**Fig. 3 Prioritize patients by disease mixture from MIMIC-III dataset. a** Word clouds of the three topics with colors indicating different data types and font size proportional to the probabilities of the EHR codes under that topic. **b** We selected top 50 patients under three of the learned topics for illustration. These topics are displayed as word clouds where the size of the word is proportional to their probabilities under the topic. The heatmap indicates patient mixture memberships for the topics. The color bar on the right of the heatmap indicates the actual diagnoses of the diseases related to each of the topics, namely leukemia, pulmonary embolism, and cirrhosis. **c** Top EHR codes of the high-risk patients. The top ten EHR codes (rows) from each topic were displayed for the top 50 patients (columns) under each topic. We highlighted some evidence (asterisks) as why some of the patients in absence of ICD-9 codes were prioritized as high risk by our method based on their relevance to the diseases of interests (i.e., leukemia, pulmonary embolism, and cirrhosis).

We further investigated the differences between the M19 and M7 topics in terms of the underlying 7731 EHR codes. For each EHR code, we grouped the patients based on the presence and absence of that code. We then tested whether the two groups are significantly different in terms of their topic mixture memberships under M19 or M7 using Wilcoxon signed-rank one-sided tests. For the ease of interpretability, here we tested whether the patients with the code exhibit significantly higher topic mixture for M19 or M7, to assess positive enrichment for the topic. We note that these phenome-wide association studies (PheWAS) results were not corrected for multiple testing, and thus we used the results as an exploratory purpose only.

While a large proportion of the significant EHR codes is associated with both the M7 and M19 topics, we also find interesting codes that are unique to each topic (Fig. 5). For instance, ICD-9 codes for suicidal ideation (V62.84), family history of psychiatric condition (V17.0), and bipolar type I disorder with the most recent severe depressed episode without psychotic behavior (296.53) are significantly associated with M19 but not M7, although both topics share 296.80 (bipolar disorder, unspecified) and 296.52 (bipolar I disorder, most recent episode (or current) depressed, moderate) codes for BD (Fig. 5). Certain lab tests (e.g., SLC6A4, HTR2A, and cytochrome P450 enzymes) are also associated with M19, but not M7. Lithium lab test is also associated with high M19 mixture probabilities, but not M7. Thus it is possible that M19 patients may have had more severe symptoms and needed an increased use of pharmacogenomically guided treatment. PheWAS for the PPI questions are displayed in Supplementary Fig. 32. Interpretation on the PPI PheWAS must be taken with caution ("Methods"). With these data, we are able to demonstrate a potential utility of our MixEHR approach to classify BD patients into potentially clinically meaningful categories that require further investigation in larger dataset.

**EHR code prediction in MIMIC-III data.** Retrospective EHR code prediction has its value in diagnosing the code entered in the existing EHR patient records and making suggestions about the potential missing code and incorrectly entered code based on the expectation of those EHR codes. To predict EHR codes, we formulated a $k$-nearest neighbor approach (Supplementary Fig. 16a; "Methods"). We evaluated the prediction accuracy based on fivefold cross-validation. The predicted EHR codes match consistently with the observed frequency of the diagnostic codes (Supplementary Fig. 15). Overall, the median prediction accuracy of MixEHR (multimodal) is 88.9%, AUROC is 85%, and AUPRC is 9.6%. In contrast, the baseline model that ignores distinct data types (flattened; i.e., LDA) obtained only 83.3% accuracy, 77.1% AUROC, and 6.7% AUPRC, respectively (Supplementary Fig. 16b). The prediction performances vary among distinct data types (Supplementary Fig. 17) and also among different ICD-9 disease groups (Supplementary Fig. 18), which are attributable to the rareness of the diseases, complexity of the disease codes, and the strength and limitations of our model-based assumption. One caveat in this experiment is that we are potentially using highly related code to predict the current target code. Therefore, we should distinguish this experiment from predicting diagnostic code in the future admissions of inpatient data or future visits in the outpatient data.

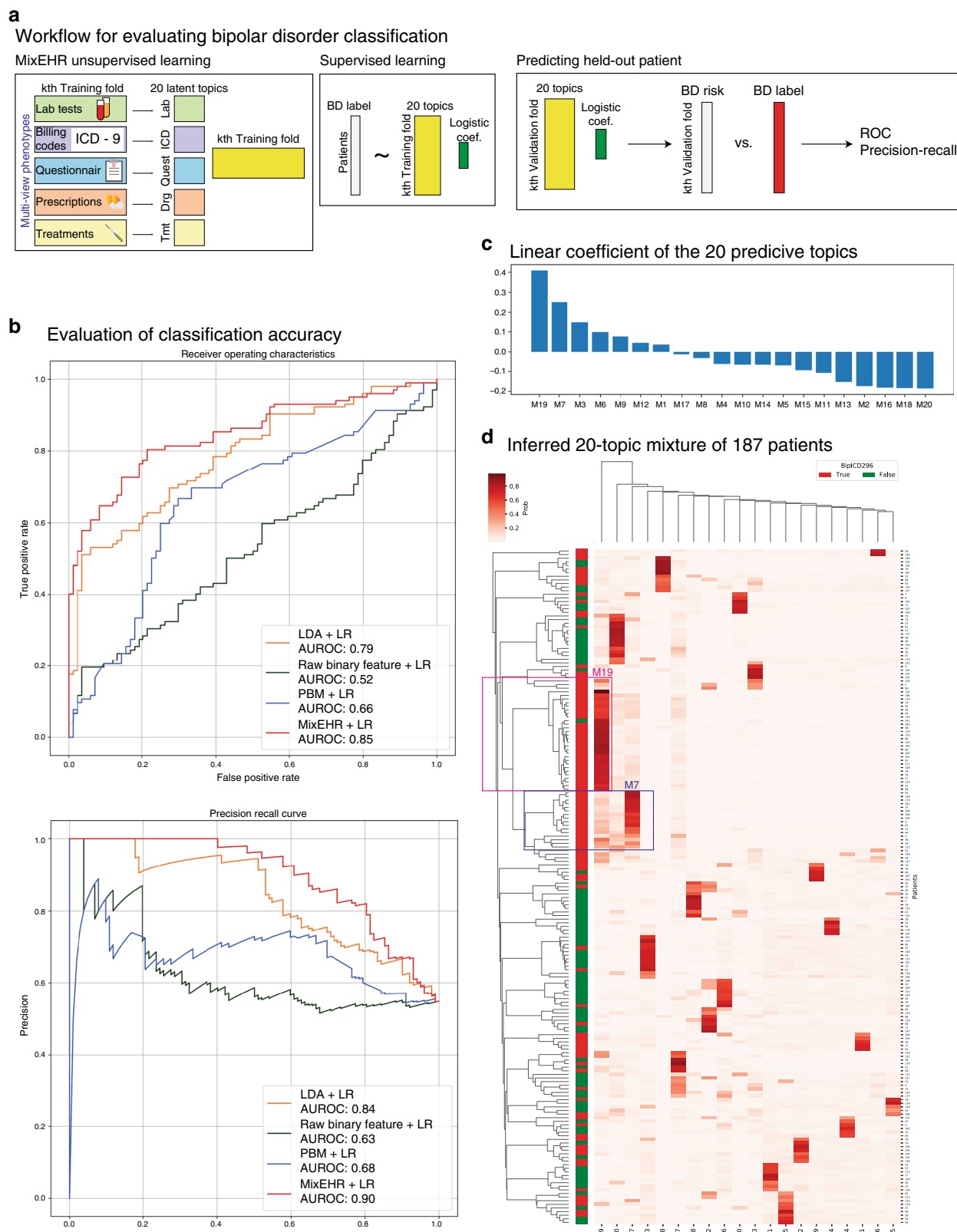

**Fig. 4 Classification of bipolar disorder patients using the Mayo Clinic dataset. a** Workflow to evaluate the classification accuracy of bipolar disorder in a fivefold cross-validation (CV). **b** Classification accuracy of fivefold CV in ROC and precision–recall curves. **c** Linear coefficients of the 20 topics ranked by decreasing order. **d** Inferred 20-topic mixture of 187 patients. The patients with and without bipolar disorder diagnosed were colored in red and green, respectively, on the left side of the dendrogram.

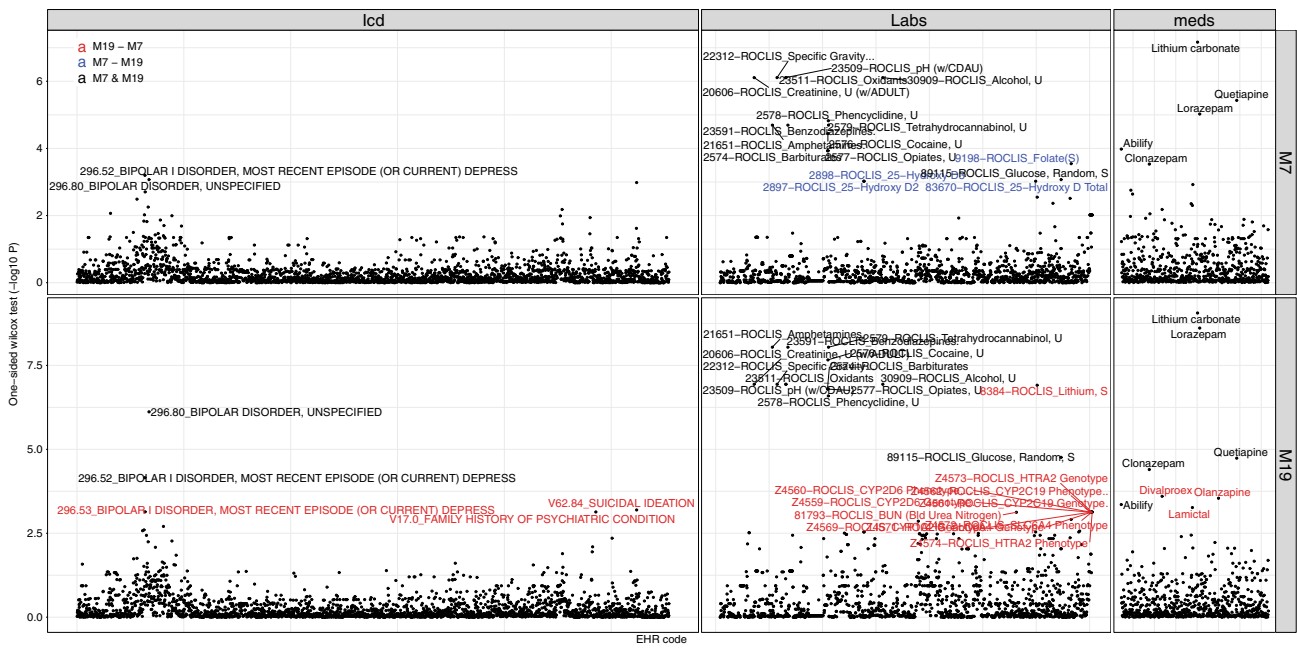

**Fig. 5 Phenome-wide association studies (PheWAS) of the 6776 EHR codes on the two bipolar disorder mixture topics (M7 and M19) based on the Mayo Clinic dataset.** For each EHR code, we tested the difference between patient groups with and without that code in terms of M19 and M7 topic membership probabilities. The labeled EHR codes are the ones with Wilcoxon signed-rank one-sided test p-values < 0.001. The red, blue, and black color indicate the codes that are significant in only M19, in only M7, and in both M19 and M7, respectively. PheWAS for the PPI questions are displayed in Supplementary Fig. 32.

In the MIMIC-III data, there are 7541 patients who were admitted to the hospital more than once. This allows us to evaluate how well we can predict the diagnostic codes in the next admission based on the information from the previous admission(s) for the same patients. To this end, we developed a pipeline that combines MixEHR topics with recurrent neural network (RNN) with Gated Recurrent Unit (GRU) (Fig. 6a; "Methods"). By leveraging the correlation structure information between rarely observed codes and commonly observed codes via the latent topics, we hypothesize that our approach can achieve comparable prediction accuracy as GRAM, which relies on the accuracy of the existing ICD-9 knowledge graph or taxonomy.

We observe that MixEHR+RNN confers the highest overall accuracy among the three methods, although the difference between MixEHR+RNN and GRAM[32] is small (Fig. 6b). We also generated the prediction accuracy on the codes binned by their frequencies as before (Supplementary Fig. 35). MixEHR+RNN performs the best in four out of the five ranges and falls short by only 2% in the last range (80–100) compared with GRAM. Both MixEHR+RNN and GRAM outperform Doctor AI[27] by a large margin. Doctor AI did not do well in this application because of the small sample size and short medical history of the MIMIC-III data (11 years) (Fig. 6b).

**Longitudinal EHR code prediction using Quebec CHD data.** We evaluated the code prediction accuracy on the 28-year longitudinal outpatient data from the CHD database using a MixEHR +RNN architecture (Fig. 6c; "Methods"). Compared to our model with the baseline RNN, we observed a significant improvement in terms of both AUPRC (Wilcoxon signed-rank tests p-value < 0.0408) and AUROC (Wilcoxon signed-rank tests p-value < 5.35e-55) (Fig. 6d). Therefore, adding MixEHR significantly improves the EHR code prediction over the baseline RNN model. We also observed that the learned weights connected to the 50-topic mixture exhibit higher magnitude than the learned weights connected to the concatenated dense layer embedding

(Supplementary Fig. 19). This means that the network relies heavily on the topic mixture to make accurate predictions. Because our dataset is focused on the CHD patients, we checked the prediction accuracy on ICD-9 code 428. Both models achieve 93% AUROC and 33% AUPRC for predicting 428 ICD-9 code.

We discovered two interesting topics namely M43 and M1 that are highly related to heart-failure ICD-9 code 428 (Supplementary Fig. 20). Specifically, M43 involves not only ICD-9 code 428.9 for heart failure but also code 518.4 for acute lung edema, code 290.9 for senile psychotic condition, code 428.0 for congenital heart failure, code 402.9 for hypertensive heart disease without HF, and code 782.3 for edema. Indeed, edema is known to be the precursor for heart failure among many patients. Interestingly, topic M1 characterizes a different set of heart-related diseases, such as rheumatic aortic stenosis (code 395.0), secondary cardiomyopathy (code 425.9), and left heart failure (code 428.1).

Procedural codes 1-71, 1-9115, 1-9112 all indicate billing for more complex care either by a family doctor or specialists in patients who have impaired mobility, prolonged hospitalization beyond 15 days and/or admission to short-stay units for patients unable to go home, typically requiring diuretics for pulmonary edema. Therefore, most of the top EHR codes under topic M43 are clinically relevant predictors of heart-failure events in clinical practice.

**Lab results imputation using MIMIC-III data.** We then evaluated the imputation accuracy of missing lab results. We first performed an extensive simulation by subsampling from MIMIC-III data to compare lab imputation accuracy between MixEHR and MICE[34] (Supplementary Note). Our approach demonstrated superior performance compared with MICE and the model that assumes missing-at-random (MAR) lab results (Supplementary Fig. 25). We then demonstrated our approach on imputing the real MIMIC-III lab results. Here, we took a similar approach as the k-nearest neighbor approach in the EHR code prediction (Fig. 7a; "Methods"). We observed that MixEHR achieves

**Fig. 6 Diagnostic code prediction using the MIMIC-III and Quebec CHD datasets. a** Proposed MixEHR+RNN framework for predicting medical code from the MIMIC-III data. **b** Medical code-prediction accuracy. We evaluated the prediction accuracy of the 42 common Clinical Classification Software (CCS) medical codes comparing MixEHR with Doctor AI and GRAM. For each medical code, the accuracy was calculated by the true positives among the top 20 predicted codes divided by the total number of positives. The dots are individual accuracies for each code predicted by each method. **c** Proposed MixEHR +RNN framework for predicting the ICD code on Quebec CHD database. **d** Prediction accuracies in terms of area under the precision–recall curve (AUPRC) and area under the ROC curve (AUROC) over all of the 1098 3-digit ICD codes. The center line, bounds, and whiskers of the boxplots are median, first and third quartiles, and outlier, respectively. Wilcoxon signed-rank one-sided tests were performed to calculate the p-values with the alternative hypothesis set to that AUCs from MixEHR+RNN are greater than the AUCs from the baseline RNN.

significantly higher accuracy compared to CF-RBM (Fig. 7b, c; Wilcoxon test $p$-value $< 0.00013$, Kolmogorov–Smirnov (KS) test $p$-value $< 1.15 \times 10^{-5}$). This is attributable to two facts: (1) Mix-EHR accounts for NMAR by jointly modeling the distribution both the lab missing indicators and lab results; (2) MixEHR is able to leverage more information than CF-RBM: it models not only the lab data but also other administrative data and clinical notes. We also observed significantly higher accuracy for our MixEHR imputation strategy compared to a simple averaging approach (Supplementary Fig. 34; "Methods").

**Mortality-risk prediction using the MIMIC-III data.** Predicting patient's mortality risk in the Intensive Care Unit (ICU) is vital to assessing the severity of illness or trauma and facilitating subsequent urgent care. Because we are able to distill meaningful disease topics from heterogeneous EHR data, we sought to assess the predictive capacity of these disease topics for mortality predictions in future admissions. Here, we predicted the mortality outcome in the last admission based on the second-last admission of the patient within 6 months. We obtained the highest 31.62% AUPRC from MixEHR (50 topics) among all methods (Fig. 8a)

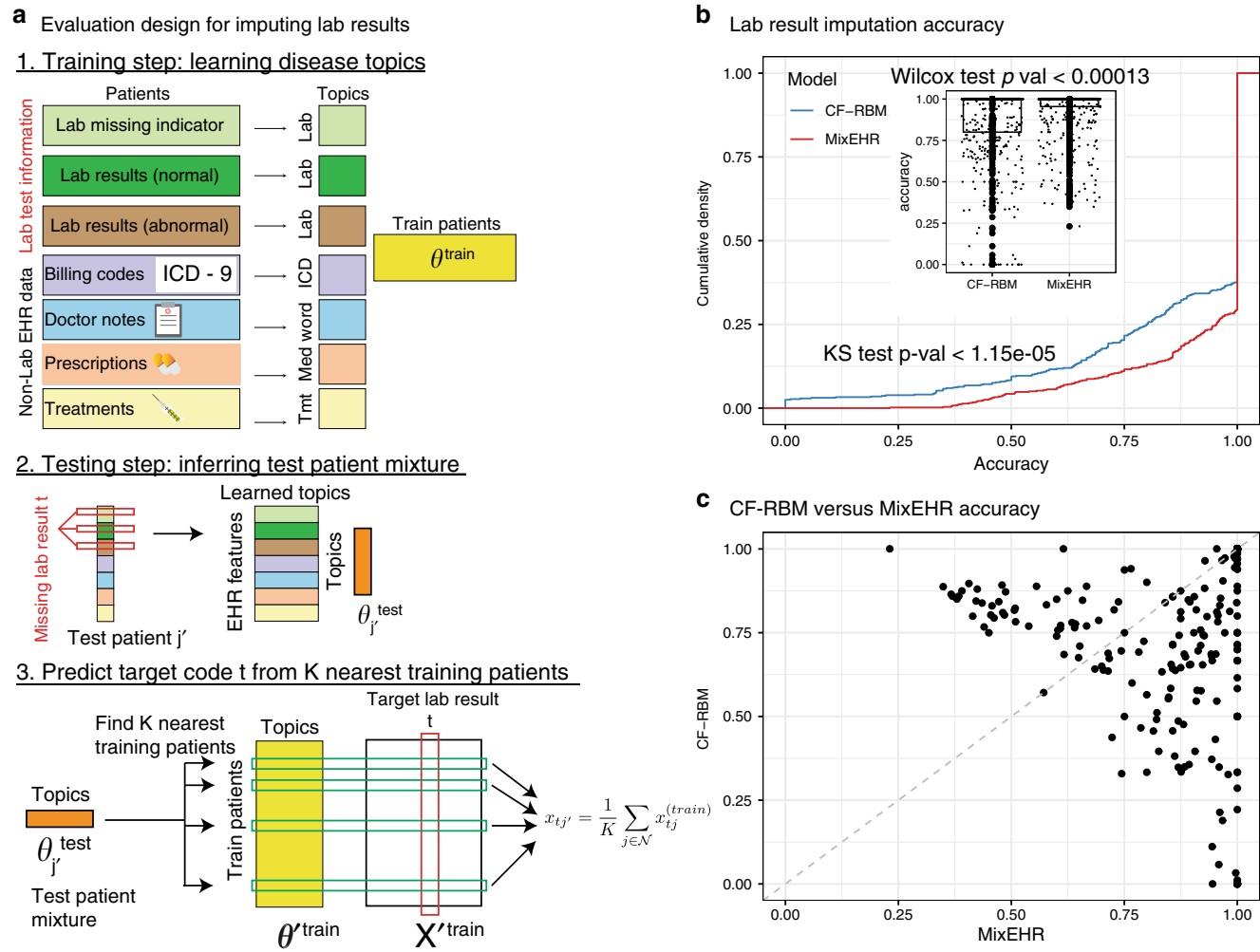

**Fig. 7 Lab results imputation using MIMIC-III dataset. a** Workflow to impute lab results. Step 1: We modeled lab tests, lab test results, and non-lab EHR data (i.e., ICD, notes, prescription, treatment) to infer the patient topic mixture. Step 2: For a test patient, we masked each of his observed lab test result $t$, and inferred his topic mixture. Step 3: We then found $k = 25$ patients who have the lab test results $t$ observed and exhibit the most similar topic mixture to the test patient. We then took the average of lab result values over the $k$ patients as the prediction of the lab result value for the test patient $j'$. Steps 1–3 were repeated to evaluate every observed lab test in every test patient. **b** We compared the imputation accuracy between MixEHR and CF-RBM. We generated the cumulative density function (CDF) of accuracy as well as the boxplot distributions (inset) for each method. The center line, bounds, and whiskers of the boxplots are median, first and third quartiles, and outlier, respectively. In both cases, MixEHR significantly outperformed CF-RBM based on KS test ($p < 1.15e-5$) and Wilcoxon signed-rank one-sided test ($p < 0.00013$). **c** CF-RBM versus MixEHR scatterplot in terms of imputation accuracy.

and second highest AUROC of 65.69% (Supplementary Fig. 22). We also experimented with two different classifiers namely logistic regression with L2-norm (Supplementary Fig. 29) and random forest (Supplementary Fig. 30). We observed overall best performance using our MixEHR embedding with both classifiers compared with the other methods.

Furthermore, we designed another experiment that used the earliest/first admission to predict the mortality in the last admission. The value of this application is to identify patients who are discharged too early and therefore to provide a measure on whether the patients should be discharged from the ICU based on their current condition. Once again, MixEHR outperformed other baseline methods (Supplementary Fig. 33). Here, we focused our analysis on the mortality in the last admission based on the information in the second-last admission.

Notably, LDA obtained similar performance to MixEHR, whereas PCA, SiCNMF, and EN performed a lot worse on this task. This

suggests that the topic models (i.e., LDA and MixEHR) are generally more suitable to modeling the discrete count data in the EHR. Based on the elastic net linear coefficients, the three most positively predictive mortality topics are enriched for renal failure (M60 with the most positive coefficient), leukemia (M73), and dementia (M26), respectively (Fig. 8c). Interestingly, the three most negatively predictive mortality topics are normal newborn (topic M52 with the most negative coefficient), aneurysm (M8), drug poisoning (M73), respectively. We note that each MixEHR topic is represented by the top EHR features from diverse data types. In contrast, all of the predictive topics from LDA are represented by a single data type: clinical notes (Fig. 8d). This is because clinical notes contain most EHR features (i.e., words) among all of the data types. By normalizing the EHR features across all of the data types, the LDA topics are overwhelmed by the category that contains a lot more features than the other categories. Qualitatively, this greatly reduces the interpretability of the topics compared to our MixEHR.

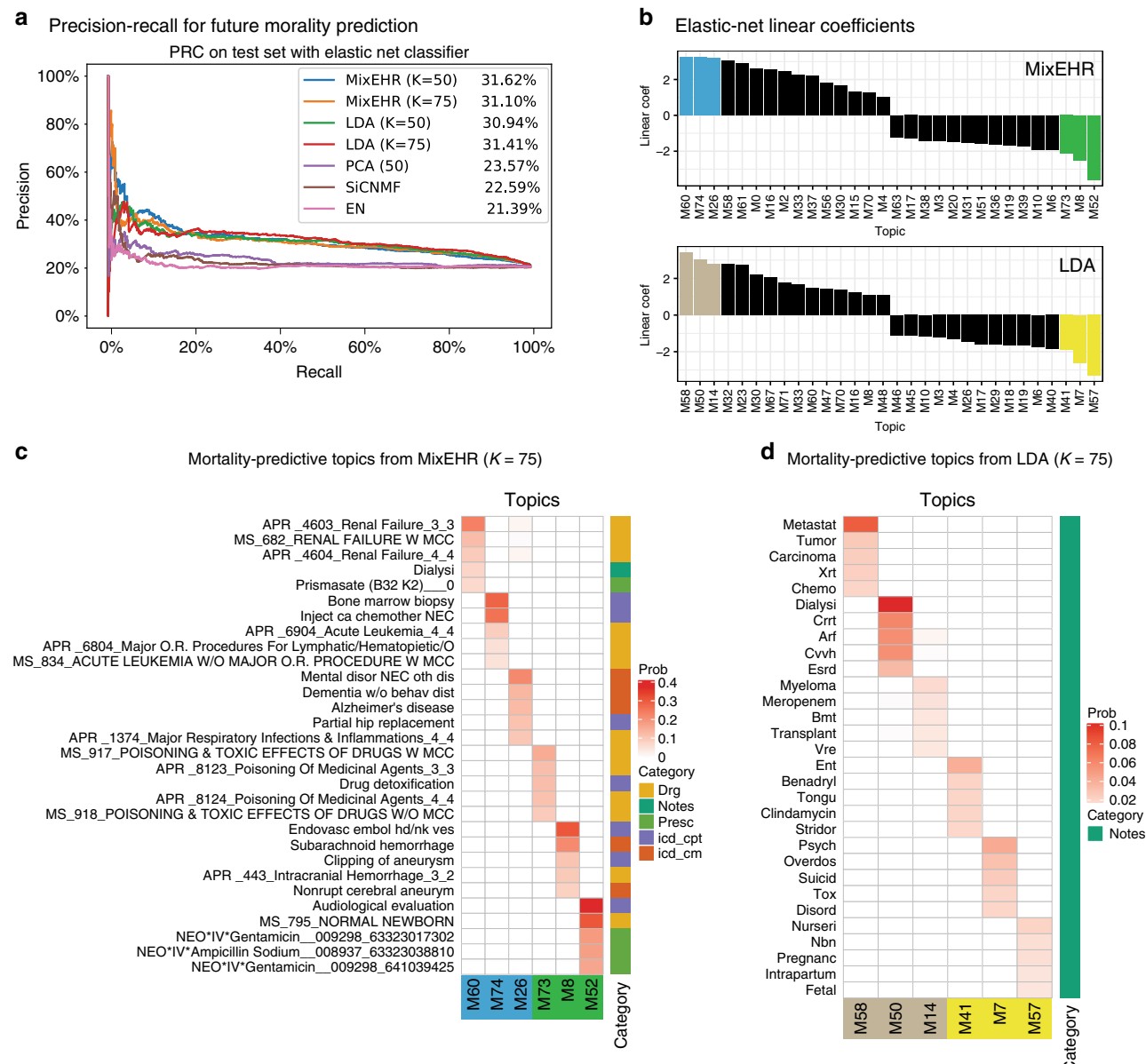

**Fig. 8 Mortality prediction using MIMIC-III dataset.** Each unsupervised embedding method was trained on the patients with only one admission in the MIMIC-III data. The trained model was then applied to embed the second-last admission from patients with at least two admissions that are within 6 months apart. An elastic net (EN) classifier was trained to predict the mortality outcome in the last admission. This was performed in a fivefold cross-validation setting. **a** Precision–recall curve (PRC) were generated, and the area under of the PRC (AUPRC) were displayed in the legend for each embedding method. EN represents the performance of elastic net using the raw EHR features. **b** Linear coefficients for the topics from MixEHR and LDA. The top three and bottom three topics are highlighted. **c** Topic clinical features for the top three most positively predictive topics and three most negatively predictive topics based on the elastic net coefficients for the 75 latent disease topics from MixEHR. **d** Same as **c**, but for the top predictive topics from LDA.

We also ran Deep Patient[29] on the MIMIC-III data and did not obtain good performance ("Methods"; Supplementary Fig. 23). This is perhaps due to the small sample size and the lack of carefully designed preprocessing pipeline[29] as the authors demonstrated on their own EHR data. In addition, previous work has shown the benefits of using demographic information in representing EHR codes and patient visits[35]. Our preliminary results show little benefit of adding demographic information including sex, age, religion, ethnicity, and marital status in the mortality prediction (Supplementary Fig. 24). However, we acknowledge its importance and will continue exploring this in our future work.

## Discussion

We present MixEHR as an unsupervised Bayesian framework to efficiently model the distribution of heterogeneous high-dimensional EHR data. We envision MixEHR to be a step toward several promising translational venues, such as (1) facilitating clinicians' decision-making, (2) directing further/deeper phenotyping, (3) enabling personalized medical recommendations, and (4) aiding biomarker discovery. Specifically, MixEHR can infer the expected phenotypes of a patient conditioned only on a subset of clinical variables that are perhaps easier and cheaper to measure. Because new data are constantly entering the

EHR systems, our model needs to adapt to incoming data without refitting to the entire dataset—in other words, to go online—a topic for future development.

In the current framework, MixEHR assumes that the input data are a set of two-dimensional matrices of patients by measurements (lab tests, diagnoses, treatments, etc.), as delineated in the structure of our datasets. However, one could envision EHR data being represented as higher-dimensional data objects—for instance, as a three-dimensional object of patients by lab tests by diagnoses, patients by diagnoses by treatments, or patients by diagnoses by time. To model such high-dimensional EHR data (as it becomes available), we can extend MixEHR to a probabilistic tensor-decomposition (TD) framework. Recently developed non-Bayesian models such as Marble[36], Rubik[37], and non-negative TD[21] are TD-based methods that show great promise. MixEHR is an ideal modeling core to serve as the basis for such upcoming modeling high-dimensional challenges.

It is challenging to interpret associations with patient provided information (PPI), compared with the other data types we used here. The presence of PPI data is confounded by factors such as the point of care (e.g., primary care versus psychiatric department), gender, demographics, and other social determinants of health. We will revisit this challenge in our future work.

Our basic MixEHR framework is cross-sectional. Nonetheless, longitudinal EHR learning is an important research area as it promises to forecast patients' disease outcomes based on their entire medical history records while taking into account temporal trajectory of patient states[27,38,39]. To this end, we present a pipeline that provides the MixEHR-inferred topic mixture at each admission as an input to a recurrent neural network in order to model the longitudinal aspect of the EHR data. It is more challenging to model irregularly sampled time points as commonly observed in the outpatient visits while modeling high-dimensional variables. In addition, in our lab imputation analysis, we did not model the sequential event of lab test results within the same admission, which requires a more sophisticated model that takes into account NMAR and the irregularly measured lab tests during the patient's stay. We leave these as future extensions to MixEHR.

In the mortality prediction, we used the information at the early admission to predict mortality in the last admission. Meanwhile, we understand the value of using the first 24 or 48 h data to predict in-hospital mortality within the same admission (e.g. ref. [17]). This is different from our application. We did not carry out this experiment because not all of the EHR data in MIMIC-III have a time stamp (i.e., CHARTTIME) within the same admission. In particular, although clinical notes and lab tests have chart time that records the time they were taken during the in-hospital admission, all of the rest of the data including ICD-9 diagnostic code, prescription, CPT, DRG code do not have a chart time and are usually entered upon patient's discharge. This makes it difficult to design such an experiment that uses all of the information. We acknowledge that one caveat in our analysis is that some patients who are terminally ill (e.g., at the metastatic cancer stage) may choose to die at home and therefore our prediction may reflect not only the patients' health states but also their mental and social states.

In summary, our data-driven approach, using a large number of heterogeneous topics over long observation periods, has the potential of leading to clinically meaningful algorithms that will help clinicians anticipate high disease-burden events in clinical practice. Our findings can inform the growing emphasis on proactive rather than reactive health management as we move into an era of increasing precision medicine.

## Methods

**Mixture EHR topic model (MixEHR).** We model EHR data using a generative latent topic model (Fig. 1). Notations are summarized in Supplementary Table 3. Suppose there are $K$ latent disease topics, each topic $k \in \{1, \ldots, K\}$ under data type $t \in \{1, \ldots, T\}$ prescribes a vector of unknown weights $\boldsymbol{\phi}_k^{(t)} = [\phi_{wk}^{(t)}]_{W^{(t)}}$ for $W^{(t)}$ EHR features, which follows a Dirichlet distribution with unknown hyperparameter $\beta_{wt}$. In addition, each topic is also characterized by a set of unknown Dirichlet-distributed weights $\boldsymbol{\eta}_{lk} = [\eta_{lkv}]_{V_l}$ with $V_l$ distinct values (e.g., a lab test $l$ with $V_l$ distinct lab result values). For each patient $j \in \{1, \ldots, D\}$, the disease mixture membership $\boldsymbol{\theta}_j$ is generated from the $K$-dimensional Dirichlet distribution $Dir(\boldsymbol{\alpha})$ with unknown asymmetric hyperparameters $\alpha_k$. To generate EHR observation $i$ for patient $j$, a latent topic $z_{ij}^{(t)}$ under data type $t$ is first drawn from multinomial distribution with rate $\boldsymbol{\theta}_j$. Given the latent topic $z_{ij}^{(t)}$, a clinical feature $x_{ij}^{(t)}$ is drawn from multinomial distribution with rate equal to $\boldsymbol{\phi}_{z_{ij}^{(t)}}^{(t)}$.

For lab data, we use a generative process where each patient has a variable for the result for every lab test regardless whether the results is observed. One important assumption we make here is that the lab results $y_{lj}$ and lab observation $r_{lj}$ for patient $j$ are conditionally independent given the latent topic $h_{lj}$, namely that the probability of taking the test and the value of that test both depend on the latent disease topic, but not on each other. In terms of the generative process, we first sample the latent variable $h_{lj}$ from the multinomial distribution with rate $\boldsymbol{\theta}_j$. Conditioned on the latent variable, we then concurrently sample (1) the lab result $y_{lj}$ from $V_l$-dimensional Dirichlet distribution $\boldsymbol{\eta}_{lh_{lj}}$ with hyperparameters $\boldsymbol{\zeta}_l$ over $V_l$ values and (2) the lab observation indicator $r_{lj}$ from Binomial distribution[1] with rate $\psi_{lh_{lj}}$, which follows a Beta distribution with unknown shape and scale hyperparameters $a_l$ and $b_l$. The hyperparameters $\alpha_k, \beta_{wt}, \zeta_{lv}, a_l, b_l$'s follow unknown Gamma distributions.

Formally, we first generate global variables as the parameters for the disease topic distribution:

$$\boldsymbol{\phi}_k^{(t)} \sim Dir(\boldsymbol{\beta}_t) : \frac{\Gamma(\sum_w \beta_{wt})}{\prod_w \Gamma(\beta_{wt})} \prod_w \left[\phi_{wk}^{(t)}\right]^{\beta_{wt}-1}$$

$$\boldsymbol{\eta}_{lk} \sim Dir(\boldsymbol{\zeta}_l) : \frac{\Gamma(\sum_v \zeta_{lv})}{\prod_v \Gamma(\zeta_{lv})} \prod_v \eta_{lkv}^{\zeta_{lv}-1}, \quad \psi_{lk} \sim Bet(a_l, b_l) : \frac{\Gamma(a_l + b_l)}{\Gamma(a_l)\Gamma(b_l)} \psi_{lk}^{a_l-1} (1-\psi_{lk})^{b_l-1}$$

We then generate local variables for the diagnostic codes including words from clinical notes, ICD-9 diagnostic codes, prescription terms, Current Procedure Terminology (CPT) codes of each patient:

$$\boldsymbol{\theta}_j \sim Dir(\boldsymbol{\alpha}) : \frac{\Gamma(\sum_k \alpha_k)}{\prod_k \Gamma(\alpha_k)} \prod_k \theta_{jk}^{\alpha_k-1}; \quad z_{ij} \sim Mul(\boldsymbol{\theta}_j) : \prod_k \theta_{jk}^{[z_{ij}=k]};$$

$$x_{ij} \sim Mul(\boldsymbol{\phi}_k^{(t)}) : \prod_w \phi_{kw}^{[x_{ij}=w]}$$

and the local variables for the lab data, including latent topic, lab result, and observation indicator:

$$h_{lj} \sim Mul(\boldsymbol{\theta}_j) : \prod_k \theta_{jk}^{[h_{lj}=k]}; \quad y_{lj} \sim Mul(\boldsymbol{\eta}_{lh_{lj}}) : \prod_v \eta_{lh_{lj}v}^{y_{ljv}};$$

$$r_{lj} \sim Bin(\psi_{lh_{lj}}) : \psi_{lh_{lj}}^{r_{lj}} (1 - \psi_{lh_{lj}})^{1-r_{lj}}$$

where $Gam(.)$, $Dir(.)$, $Bet(.)$, $Mul(.)$, and $Bin(.)$ denote gamma, Dirichlet, beta, multinomial, and Binomial distributions, respectively.

**Marginalized likelihood.** Treating the latent variables as missing data, we can express the complete joint likelihood based on our model as follows:

$$p(\mathbf{x}, \mathbf{y}, \mathbf{r}, \mathbf{z}, \mathbf{h}, \boldsymbol{\theta}, \boldsymbol{\phi}, \boldsymbol{\psi}, \boldsymbol{\eta} | \boldsymbol{\Theta}) = p(\boldsymbol{\theta}|\boldsymbol{\alpha}) p(\mathbf{z}, \mathbf{h}|\boldsymbol{\theta}) p(\mathbf{x}|\mathbf{z}, \boldsymbol{\phi}) p(\boldsymbol{\phi}|\boldsymbol{\beta}) p(\mathbf{r}|\mathbf{h}, \boldsymbol{\psi}) p(\mathbf{y}|\mathbf{h}, \boldsymbol{\eta}) p(\boldsymbol{\eta}|\boldsymbol{\zeta}) p(\boldsymbol{\psi}|\mathbf{a}, \mathbf{b})$$

At this point, we could formulate a variational inference algorithm with mean-field factorization by optimizing an evidence lower bound of the marginal likelihood with respect to the model parameters[40]. However, that would be inaccurate and computationally expensive for the following reasons. The mean-field variational distribution assumes that all of the latent variables are independent. On the other hand, we know from our model that these independence assumptions are unrealistic. In particular, the latent topic assignments depend on the patient mixtures, and the disease latent topic distribution and topic assignment are dependent given the observed lab tests or diagnostic code (i.e., the V shape structure in the probabilistic graphical model[41]). In addition, the variational updates require extensive usages of exponential function and digamma function for every latent topic assignment of every patient, which are computationally expensive to compute for larger number of patients.

We can achieve more accurate and efficient inference by first analytically integrating out the patient mixtures and latent topic distributions[13,42]. To do so, we exploit the respective conjugacy of Dirichlet variables $\boldsymbol{\phi}, \boldsymbol{\eta}, \boldsymbol{\theta}$ to the multinomial likelihood variables $\mathbf{x}, \mathbf{y}, \{\mathbf{z}, \mathbf{h}\}$, and the conjugacy of Beta variable $\boldsymbol{\psi}$ to the binomial lab observation indicator variable $\mathbf{r}$. This way we can simplify the complete joint

likelihood distribution of the data and the latent variables $\{\mathbf{z}, \mathbf{h}\}$ as follows:

$$p(\mathbf{x}, \mathbf{y}, \mathbf{r}, \mathbf{z}, \mathbf{h}|\boldsymbol{\alpha}, \boldsymbol{\beta}, \boldsymbol{\zeta}, \mathbf{a}, \mathbf{b}) = \int p(\mathbf{z}, \mathbf{h}|\boldsymbol{\theta})p(\boldsymbol{\theta}|\boldsymbol{\alpha})d\boldsymbol{\theta} \int p(\mathbf{x}|\mathbf{z}, \boldsymbol{\phi})p(\boldsymbol{\phi}|\boldsymbol{\beta})d\boldsymbol{\phi}$$

$$\int p(\mathbf{y}|\mathbf{h}, \boldsymbol{\eta})p(\boldsymbol{\eta}|\boldsymbol{\zeta})d\boldsymbol{\eta} \int p(\mathbf{r}|\mathbf{h}, \boldsymbol{\psi})p(\boldsymbol{\psi}|\mathbf{a}, \mathbf{b})d\boldsymbol{\psi}$$

$$= \prod_j \frac{\Gamma(\sum_k \alpha_k)}{\prod_k \Gamma(\alpha_k)} \frac{\prod_k \Gamma\left(\alpha_k + n_{.jk}^{(.)} + m_{jk}\right)}{\Gamma\left(\sum_k \alpha_k + n_{.jk}^{(.)} + m_{jk}\right)} \prod_k \prod_t \frac{\Gamma(\sum_w \beta_{wt})}{\prod_w \Gamma(\beta_{wt})} \frac{\prod_w \Gamma\left(\beta_t + n_{w.k}^{(t)}\right)}{\Gamma\left(\sum_w \beta_{wt} + n_{w.k}^{(t)}\right)}$$

$$\prod_k \prod_l \frac{\Gamma(\sum_v \zeta_{lv})}{\prod_v \Gamma(\zeta_{lv})} \frac{\prod_v \Gamma(\zeta_{lv} + m_{l.kv})}{\Gamma(\sum_v \zeta_{lv} + m_{l.kv})} \prod_k \prod_l \frac{\Gamma(a_l + b_l)}{\Gamma(a_l)\Gamma(b_l)} \frac{\Gamma(a_l + p_{lk})\Gamma(b_l + q_{lk})}{\Gamma(a_l + p_{lk} + b_l + q_{lk})}$$

where the sufficient statistics are:

$$n_{.jk}^{(.)} = \sum_{t=1}^T \sum_{i=1}^{M_j^{(t)}} [z_{ij}^{(t)} = k], \quad n_{w.k}^{(t)} = \sum_{j=1}^D \sum_{i=1}^{M_j^{(t)}} [x_{ij}^{(t)} = w, z_{ij}^{(t)} = k] \quad (1)$$

$$m_{.jk.} = \sum_{l=1}^L \sum_{v=1}^{V_l} y_{ljv}[h_{lj} = k], \quad m_{l.kv} = \sum_{j=1}^D y_{ljv}[h_{lj} = k] \quad (2)$$

$$p_{lk} = \sum_j [r_{lj} = 1] \prod_v y_{ljv}[h_{lj} = k], \quad q_{lk} = \sum_j [r_{lj} = 0] \sum_v [h_{lj} = k, y_{lj} = v] \quad (3)$$

Note that we use $y_{ljv}$ to denote the frequency of observing the lab test $l$ of outcome $v$ for patient $j$ and $[y_{lj} = v]$ as binary indicator of a single test. Detailed derivations are in Supplementary Methods.

**Joint collapsed variational Bayesian inference**. The marginal likelihood can be approximated by evidence lower bound (ELBO):

$$\log p(\mathbf{x}, \mathbf{y}, \mathbf{r}|\boldsymbol{\alpha}, \boldsymbol{\beta}, \boldsymbol{\zeta}, \mathbf{a}, \mathbf{b}) = \log \sum_{\mathbf{z}, \mathbf{h}} \frac{p(\mathbf{x}, \mathbf{y}, \mathbf{r}, \mathbf{z}, \mathbf{h}|\boldsymbol{\alpha}, \boldsymbol{\beta}, \boldsymbol{\zeta}, \mathbf{a}, \mathbf{b})}{q(\mathbf{z}, \mathbf{h})} q(\mathbf{z}, \mathbf{h}) \quad (4)$$

$$\geq \sum_{\mathbf{z}, \mathbf{h}} q(\mathbf{z}, \mathbf{h}) \log p(\mathbf{x}, \mathbf{y}, \mathbf{r}, \mathbf{z}, \mathbf{h}|\boldsymbol{\alpha}, \boldsymbol{\beta}, \boldsymbol{\zeta}, \mathbf{a}, \mathbf{b}) - \sum_{\mathbf{z}, \mathbf{h}} q(\mathbf{z}, \mathbf{h}) \log q(\mathbf{z}, \mathbf{h}) \quad (5)$$

$$= \mathbb{E}_{q(\mathbf{z}, \mathbf{h})}[\log p(\mathbf{x}, \mathbf{y}, \mathbf{r}, \mathbf{z}, \mathbf{h}|\boldsymbol{\alpha}, \boldsymbol{\beta}, \boldsymbol{\zeta}, \mathbf{a}, \mathbf{b})] - \mathbb{E}_{q(\mathbf{z}, \mathbf{h})}[\log q(\mathbf{z}, \mathbf{h})] \equiv \mathcal{L}_{ELBO} \quad (6)$$

Eqs. (4) to (5) follows Jensen's inequality. Maximizing $\mathcal{L}_{ELBO}$ is equivalent to minimizing Kullback–Leibler (KL) divergence:

$$\mathcal{KL}[q(\mathbf{z}, \mathbf{h})||p(\mathbf{z}, \mathbf{h}|\mathbf{x}, \mathbf{y}, \mathbf{r})] = \mathbb{E}_{q(\mathbf{z}, \mathbf{h})}[\log q(\mathbf{z}, \mathbf{h})] - \mathbb{E}_{q(\mathbf{z}, \mathbf{h})}[\log p(\mathbf{z}, \mathbf{h}, \mathbf{x}, \mathbf{y}, \mathbf{r})]+$$

$\log p(\mathbf{x}, \mathbf{y}, \mathbf{r})$ because $\log p(\mathbf{x}, \mathbf{y}, \mathbf{r})$ is constant and $\mathcal{KL}[q(\mathbf{z}, \mathbf{h})||p(\mathbf{z}, \mathbf{h}|\mathbf{x}, \mathbf{y}, \mathbf{r})]+$ $\mathcal{L}_{ELBO} = \log p(\mathbf{x}, \mathbf{y}, \mathbf{r})$.

Under mean-field factorization, the proposed distribution of latent variables $\mathbf{z}$ and $\mathbf{h}$ are defined as:

$$\log q(\mathbf{z}|\boldsymbol{\gamma}) = \sum_{t,i,j,k} [z_{ij}^{(t)} = k] \log \gamma_{ijk}^{(t)}, \quad \log q(\mathbf{h}|\lambda) = \sum_{l,j,k} [h_{lj} = k] \log \lambda_{ljk} \quad (7)$$

Maximizing (6) with respect to the variational parameter $\gamma_{ijk}^{(t)}$ and $\lambda_{ljk}$ is equivalent to calculating the expectation of $z_{ij}^{(t)} = k$ and $h_{lj} = k$ with respect to all of the other latent variables[13,41]:

$$\log q(z_{ij}^{(t)} = k) = \mathbb{E}_{q(\mathbf{z}^{-(i,j)})}[\log p(\mathbf{x}, \mathbf{z})], \quad \log q(h_{lj} = k) = \mathbb{E}_{q(\mathbf{h}^{-(l,j)})}[\log p(\mathbf{y}, \mathbf{r}, \mathbf{h})] \quad (8)$$

Exponentiating and normalizing the distribution of $q(z_{ij}^{(t)} = k)$ and $q(h_{lj} = k)$ gives

$$\gamma_{ijk}^{(t)} = \frac{\exp(\mathbb{E}_{q(\mathbf{z}^{-(i,j)})}[\log p(\mathbf{x}, \mathbf{z}^{-(i,j)}, z_{ij} = k)])}{\sum_{k'} \exp(\mathbb{E}_{q(\mathbf{z}^{-(i,j)})}[\log p(\mathbf{x}, \mathbf{z}^{-(i,j)}, z_{ij} = k')])} \quad (9)$$

$$\lambda_{ljk} = \frac{\exp(\mathbb{E}_{q(\mathbf{h}^{-(l,j)})}[\log p(\mathbf{y}, \mathbf{r}, \mathbf{h}^{-(i,j)}, h_{ij} = k)])}{\sum_{k'} \exp(\mathbb{E}_{q(\mathbf{h}^{-(l,j)})}[\log p(\mathbf{y}, \mathbf{r}, \mathbf{h}^{-(i,j)}, h_{ij} = k')])} \quad (10)$$

We can approximate these expectations by first deriving the conditional distribution for $p(z_{ij}^{(t)} = k|\mathbf{x}, \mathbf{y}, \mathbf{r})$ and $p(h_{lj} = k|\mathbf{x}, \mathbf{y}, \mathbf{r})$ (Supplementary Methods) and then using Gaussian distributions to approximate the sufficient statistics by the summation of the variational parameters in zero-order Taylor expansion[13,14]:

$$\gamma_{ijk}^{(t)} \propto \left(\alpha_k + \tilde{n}_{.jk}^{-(i,j)} + \tilde{m}_{jk}\right) \left(\frac{\beta_{tx_{ij}^{(t)}} + \left[\tilde{n}_{x_{ij}^{(t)}.k}^{(t)}\right]^{-(i,j)}}{\sum_w \beta_{wt} + \left[\tilde{n}_{w.k}^{(t)}\right]^{-(i,j)}}\right) \quad (11)$$

To infer $p(h_{lj} = k|\mathbf{x}, \mathbf{y}, \mathbf{r})$, we will need to separately consider whether the lab test $l$ is observed or missing. In particular, we can infer the topic distribution $\lambda_{ljkv} \propto \mathbb{E}_q[\log p(h_{lj} = k|\mathbf{x}, \mathbf{y}, \mathbf{r})]$ of an observed lab test $y_{lj}$ at value $v$ as:

$$\lambda_{ljkv} \propto \left(\alpha_k + \tilde{n}_{jk} + \tilde{m}_{jk}^{-(l,j)}\right) \left(\frac{\zeta_{lv} + \tilde{m}_{lkv}^{-(l,j,v)}}{\sum_{v'} \zeta_{lv'} + \tilde{m}_{lkv'}^{-(l,j,v)}}\right) \left(\frac{a_l + \tilde{p}_{lk}^{-(l,j)}}{a_l + \tilde{p}_{lk}^{-(l,j)} + b_l + \tilde{q}_{lk}}\right) \quad (12)$$

For unobserved lab tests, we infer the joint distribution of latent topic and lab result $\pi_{ljkv} \propto \mathbb{E}_q[\log p(h_{lj} = k, y_{lj} = v|\mathbf{x}, \mathbf{y}, \mathbf{r})]$:

$$\pi_{ljkv} \propto \left(\alpha_k + \tilde{n}_{jk} + \tilde{m}_{jk}^{-(l,j)}\right) \left(\frac{\zeta_{lv} + \tilde{m}_{lkv}^{-(l,j,v)}}{\sum_{v'} \zeta_{lv'} + \tilde{m}_{lkv'}^{-(l,j,v)}}\right) \left(\frac{b_l + \tilde{q}_{lk}^{-(l,j)}}{a_l + \tilde{p}_{lk} + b_l + \tilde{q}_{lk}^{-(l,j)}}\right) \quad (13)$$

where the notation $n^{-(i,j)}$ indicate the exclusion of variable index $i, j$. The sufficient statistics in the above inference have closed-form expression conditioned on the other latent variables:

$$\tilde{n}_{jk}^{-(i,j)} = \sum_{t=1}^T \sum_{i' \neq i}^{M_j^{(t)}} \gamma_{i'jk}, \quad \left[\tilde{n}_{x_{ij}^{(t)}.k}^{(t)}\right]^{-(i,j)} = \sum_{j' \neq j}^D \sum_{i=1}^{M_{j'}^{(t)}} [x_{ij}^{(t)} = x_{ij}^{(t)}] \gamma_{ij'k} \quad (14)$$

$$\tilde{m}_{.jk.}^{-(l,j)} = \sum_{l' \neq l}^L \sum_{v=1}^{V_{l'}} [r_{l'j} = 1] y_{l'jv} \lambda_{l'jkv} + [r_{l'j} = 0] \pi_{l'jkv} \quad (15)$$

$$\tilde{m}_{l.kv}^{-(l,j,v)} = \sum_{j'=1}^D \sum_{l=1}^L \sum_{v' \neq v}^{V_l} [r_{lj'} = 1] y_{ljv'} \lambda_{ljkv'} + [r_{lj'} = 0] \pi_{ljkv'} \quad (16)$$

$$\tilde{p}_{lk}^{-(l,j)} = \sum_{j' \neq j} [r_{lj'} = 1] \sum_v y_{lj'v} \lambda_{lj'k}, \quad \tilde{q}_{lk}^{-(l,j)} = \sum_{j' \neq j} [r_{lj'} = 0] \sum_v \pi_{ljkv} \quad (17)$$

Furthermore, we update the hyperparameters by maximizing the marginal likelihood under the variational expectations via empirical Bayes fixed-point updates[14,43]. For example, the update for $\beta_{wt}$ is

$$\beta_{wt}^* \leftarrow \frac{a_\beta - 1 + \beta_{wt} \left(\sum_k \Psi(\beta_{wt} + n_{w.k}^{(t)}) - KW_t \Psi(\beta_{wt})\right)}{b_\beta + \sum_k \Psi(W_t \beta_{wt} + \sum_w n_{w.k}^{(t)}) - K\Psi(W_t \beta_{wt})} \quad (18)$$

Other hyperparameters updates are similar. The learning algorithm therefore follows expectation–maximization: E-step infers $\gamma_{ijk}^{(t)}, \lambda_{ljkv}, \pi_{ljkv}$'s; M-step updates the above sufficient statistics of model parameters and hyperparameters. Details and time complexity analysis are described in Supplementary Methods.

**Stochastic joint collapsed variational Bayesian inference**. To learn MixEHR from massive-scale EHR data, we propose a stochastic collapsed variational Bayesian (SCVB) algorithm[44,45]. The main objective of our SCVB algorithm is to avoid keeping track of all of the latent topic assignments $\gamma_{ijk}^{(t)}$ and $\lambda_{ljkv}$'s while maintaining accurate model inference. Specifically, we first calculate the intermediate variational updates from randomly sampled mini-batches of $D'$ patients:

$$\hat{n}_{w.k}^{(t)} = \sum_{j'=1}^{D'} \sum_{i=1}^{M_{j'}^{(t)}} [x_{ij'}^{(t)} = w] \gamma_{ij'k}^{(t)} \quad (19)$$

$$\hat{m}_{l.kv} = \sum_{j'=1}^{D'} \sum_{l=1}^L [r_{lj'} = 1] y_{lj'v} \lambda_{lj'kv} + [r_{lj'} = 0] \pi_{lj'kv} \quad (20)$$

We then update the latent disease topics using natural gradients, which are the intermediate updates weighted by a fixed learning rate:

$$\hat{n}_{w.k}^{(t)} = (1 - \rho) \tilde{n}_{w.k}^{(t)} + \rho \hat{n}_{w.k}^{(t)} \quad (21)$$

$$\hat{m}_{l.kv}^{(t)} = (1 - \rho) \tilde{m}_{l.kv}^{(t)} + \rho \hat{m}_{l.kv}^{(t)} \quad (22)$$

Details are described in Supplementary Methods. We observed that the SJCVB0 algorithm achieves comparable performance as the full-batch learning with only a fraction of time and constant memory (Supplementary Fig. 2).

**Predictive likelihood**. To evaluate model learning and monitor empirical convergence, we performed fivefold cross-validation. For each patient in the validation fold, we randomly selected 50% of their EHR features to infer their disease mixtures and then used the other 50% of the features to evaluate the log predictive likelihood —a common metric to evaluate topic models[4,14,44]:

$$\sum_j \sum_{t,i} \log \sum_k \hat{\theta}_{jk} \hat{\phi}_{x_{ij}^{(t)}k}^{(t)} + \sum_l [r_{lj} = 1] \log \sum_k \hat{\theta}_{jk} (\hat{\psi}_{lk} + \hat{\eta}_{lky_{lj}}) \quad (23)$$

where we inferred the variational expectations of the disease mixture and disease

topics as:

$$\hat{\theta}_{jk} = \frac{\alpha_k + \tilde{n}_{jk}^{(.)}}{\sum_{k'} \alpha_{k'} + \tilde{n}_{jk'}^{(.)}}, \hat{\phi}_{wk}^{(t)} = \frac{\beta_{wt} + \tilde{n}_{wk}^{(t)}}{\sum_{w'} \beta_{w't} + \tilde{n}_{w'k}^{(t)}},$$

$$\hat{\psi}_{lk} = \frac{a_l + \tilde{p}_{lk}}{a_l + \tilde{p}_{lk} + b_l + \tilde{q}_{lk}}, \hat{\eta}_{lk\nu} = \frac{\zeta_{l\nu} + \tilde{m}_{lk\nu}}{\sum_{\nu'=1}^{V_l} \zeta_{l\nu'} + \tilde{m}_{lk\nu'}}$$

Sensible models should demonstrate improved predictive likelihood on held-out patients. We evaluated the predictive log likelihood of models with $K \in \{10, 25, 40, 50, 60, 75, 100, 125, 150\}$ using the MIMIC-III data and ultimately set **K = 75** as it gave the highest predictive likelihood (Supplementary Fig. 1). Nonetheless, models with $K$ within the range between 50 and 100 have similar performance.

**Retrospectively predict EHR codes by a k-nearest neighbor approach.** To impute missing data in an individual-specific way, we here describe a k-nearest neighbor approach. As illustrated in Supplementary Fig. 16a. the prediction can be divided into three steps:

- Train MixEHR on training set to learn the EHR code by disease topic matrices **W** across data types and infer the disease topic mixtures $\theta^{\text{train}}$ for each training patient data point;
- To infer the probability of an unknown EHR code $t$ for a test patient $j'$, use MixEHR and the learnt disease topic matrices **W** to infer the disease topic mixture $\theta_{j'}$ for the test patient;
- Compare the test patient disease topic mixture $\theta_{j'}$ with the training patient disease mixtures $\theta^{\text{train}}$ to find the $k$ most similar training patients $\mathcal{S}_{j'}$. Here, the patient–patient similarity matrix is calculated based on the Euclidean distance between their disease topic mixtures:

$$d_{ij} = \sqrt{\sum_{k=1}^{K} \left(\theta_{j'k} - \theta_{jk}\right)^2} \tag{24}$$

where $\theta_{j'k}$ and $\theta_{jk}$ are the mixture memberships of the test patient $j'$ and training patient $j$, respectively. Finally, we take the average of the EHR code $t$ over these k-nearest neighbor patients as the prediction for the target code $t$ for test patient $j'$:

$$x_{tj'} = \frac{1}{k} \sum_{j \in \mathcal{S}_{j'}} x_{tj} \tag{25}$$

We empirically determined the number of nearest neighbors $k$ to be 100.

**Evaluating retrospective EHR code prediction.** To evaluate the prediction performance, we chose a set of EHR codes from each data category based on the following criteria. For clinical notes, we chose the key words that are recorded in more than 10 patients but fewer than 500 patients to focus on disease-specific words with sufficient number of patients. For other data categories, we chose target codes that are observed in at least 100 patients but no more than 50% of the training cohort (i.e., ~19,000 patients). We then further selected the top 100 codes per data type in terms of the number of patients except for the ICD-9 diagnostic code, for which we evaluated prediction on each of the 499 codes in order to assess the performance differences by ICD-9 disease groups (Supplementary Fig. 18). In total, we obtained 976 target EHR codes (Supplementary Table 4).

For unbiased evaluation, we performed a fivefold cross-validation. For training fold $n$, we trained MixEHR to learn the disease topic matrices $W^{\text{training fold n}}$ and patient disease topic mixtures $\theta^{\text{training fold n}}$. For the corresponding validation fold, we evaluated the prediction of each target code $t$ for each test patient $j'$ as follows. We first removed the target code $t$ from the test patient $j'$ if it is observed for the test patient $j'$. We then inferred the disease mixture for the test patient $j'$ and inferred the probability of target code $t$ for test patient $j'$ using the k-nearest neighbor approach. We repeated this procedure for the five validation folds to obtain the predictions of each target EHR code.

For comparison, we evaluated the performance of the baseline topic model which ignores distinct data types in the EHR data. Notably, such model is essentially equivalent to LDA. To this end, we modified the training and validation datasets to have the same data type for all of the 53432 EHR codes and gave each code a unique ID (i.e., 1 to 53,432). We then extracted the 976 target codes for fivefold cross-validation predictions. We ran MixEHR on such data-type-flattened data. We evaluated the predictions based on area under the ROC curve (AUROC) and precision–recall curve (AUPRC) as well as the overall accuracy by thresholding the prediction probability by 1/k, where $k = 100$ is the empirically determined number of nearest neighbors.

**Implementations of the existing methods.** For experiments comparing our method with GRAM[32] and Doctor AI[27], we used the open source implementation made available by the authors of the papers found at https://github.com/mp2893/gram and https://github.com/mp2893/doctorai, respectively. GRAM and Doctor AI

were both set to predict the medical codes (as defined in their papers) in the next admission for the MIMIC-III. This task is also known as sequential diagnosis prediction. The same Clinical Classification Software (CCS) medical codes used in the original implementation of both methods were used in the comparisons as well. Note that Doctor AI can also be used to predict the time duration of next visit, but we did not compare our method with it in this aspect. For both Doctor AI and GRAM, we used the default recommended settings provided in their code bases. Both these models were implemented in Theano. To compare with Doctor AI and GRAM, we used Accuracy@20 as described in the GRAM paper for different percentile of frequencies of the medical codes.

In our comparison with Deep Patient, since there was no published code available, we implemented the stacked denoising autoencoder to the best of our abilities. We closely followed the steps mentioned in the paper by the authors. We did not follow the elaborate data preprocessing pipeline used in the paper as our dataset (MIMIC-III) is different from the one used by[29]. Our implementation had three layers of autoencoders with 500 hidden units (as in the original paper) and was coded using Keras library. We used the Deep Patient representation as input to a classifier (i.e., logistic regression) to predict mortality and compared the accuracy with MixEHR in terms of the area under the precision–recall (AUPRC) and area under the ROC (AUROC) (Supplementary Fig. 23).

Conditioned Factored Restricted Boltzmann Machine (CF-RBM) was implemented by adapting the code from https://github.com/felipecruz/CFRBM. CF-RBM was originally designed for Netflix movie recommendation. Each movie has a five-star rating. We modified the code to make it work with two-state lab results (i.e., normal and abnormal). We used 100 hidden units of CF-RBM to train on the randomly sampled 80% of the admissions and tested the CF-RBM on the remaining 20% of admissions. We ensured that the training and test sets were the same for both CF-RBM and MixEHR. We evaluated the trained CF-RBM as follows. We iterated over the admissions in the test set, and for every lab test in that admission, we masked its test result and made a prediction based on the remaining lab tests of the same admission. This procedure was repeated for every lab test in every admission. We then recorded the predicted lab results and the true lab results for evaluation purpose. The same evaluation procedure was also used for MixEHR.

SiCNMF was implemented using the Python code from author's Github repository https://github.com/sgunasekar/SiCNMF[19]. The model was trained on the MIMIC-III dataset with six data types as the six collective matrices until convergence.

Unless mentioned otherwise, all of the other unsupervised learning and classification methods such as PCA, random forest, elastic net, logistic regression, and LDA were implemented using the Scikit-Learn Python library.

**MIMIC-III data description.** The methods were performed in accordance with the PhysioNet data user agreement. All of the MIMIC-III data received simple and minimal preprocessing of the raw comma separated value (CSV) files provided from MIMIC-III critical care database (mimic.physionet.org). For all data types, we removed nonspecific EHR code that were commonly observed among admissions based on their inflection points (Supplementary Fig. 26). No further processing was performed unless mentioned otherwise. For clinical notes (NOTEEVENTS.csv), we used the R library tm[46] by following a simple pipeline. We filtered out common stop words, punctuations, numbers, whitespace and converting all of the remaining words to lowercase.

For lab test data, we used the FLAG column that indicates normal or abnormal level for the lab results. We counted for each admission the number of times the lab results were normal and abnormal. A patient at the same admission can exhibit both normal and abnormal states zero or more times for the same lab test. In this case, we recorded the frequency at each state for the same admission.

For prescription, we concatenated the DRUG, DRUG_NAME_GENERIC, GSN, NDC column to form a compound ID for each prescription. For the Diagnosis-related group (DRG) code (DRGCODES.csv), we concatenated several IDs including DRG_TYPE, DRG_CODE, DESCRIPTION, DRG_SEVERITY, and DRG_MORTALITY to form a compound ID for each DRG code. The identifiers for ICD-9 code (ICD-9-CM) (DIAGNOSES_ICD.csv) and treatment procedure code (PROCEDURES_ICD.csv) (ICD-9-CPT) were kept as their original forms. The resulting data are summarized in Supplementary Table 1.

**Mayo Clinic data description.** The 187 patients (94 bipolar disorder cases and 93 controls) were selected from the Mayo Clinic Bipolar Disorder Biobank and the Mayo Clinic Biobank, respectively. They had consented for research, including research using their EHR data[47,48]. This study was reviewed and approved by Mayo Clinic Institutional Review Board and by the access committees from both bio-banks. The EHR information for each of the patients cover five categories, including ICD-9 codes (2224 ICD-9-CM codes), procedure codes (2200 CPT codes), patient provided information (955 text questions), lab tests (1742 lab test codes), medication prescriptions (610 prescription order codes). In total, there are 7731 codes and 108,390 observations (Supplementary Table 5). For each case, one subject from the Mayo Clinic Biobank was matched on age and sex, after excluding subjects with psychiatric conditions in self-reports and/or EHR. Supplementary Table 6 summarizes age-sex distribution by cases and controls.

**Quebec CHD data description**. The dataset was derived from the EHR documented Quebec Congenital Heart Disease (CHD) database with 84,498 patients and 28 years of follow-up from 1983 to 2010[49,50]. The study uses administrative databases as data sources. No participant is enrolled. The study was approved by Research Ethics Board of McGill University Health Centre. For each patient, the patient history can be traced back to the later of birth or 1983, and followed up to the end of the study or death of the patient, whichever came earlier. The dataset includes two data types: ICD-9 or ICD-10 billing code and intervention code. The raw dataset consists of individual records of each patient's medical visit. Each visit lasted less than a day. Thus, they were not considered as admissions, and are categorized as outpatient data. The informed consent was obtained from all participants.

By using the time stamp of those visits, we aggregated the diagnoses and intervention codes by the months into visit blocks. Thus, each visit block represents the record of a patient in that time interval as a set of diagnoses and a set of interventions. Since the patients' visit history was recorded as sequential events over the 28 years, we represented each patient's complete medical history as a time series of diagnoses and intervention sets. The length of the sequence depends on the total number of months. We only kept the months in which a patient has visited a medical institution at least once. The diagnoses given were represented as a four or five-digit ICD-9 or ICD-10 code. All of the codes were converted to ICD-9 using a board-approved conversion table. In order to generate the labels, we took the first three digits of each code. The processed data are summarized in Supplementary Table 3.

**Bipolar disorder Mayo Clinic patients classification details**. Among the five data categories, lab tests and PPI have two separate data features unlike the other three binary data categories. Specifically, they consist of (1) presence vs. absence of the data in the EHR (e.g., whether a lab test was done or whether a patient answered a particular PPI question) and (2) the actual value for the variable (i.e., test results for a lab test, or patient's answer for a PPI question). Given the small sample size, we chose to model the first data feature (whether the data exists in the EHR) to mimic the data structure used in the rest of the data categories, recognizing that we may miss important information by not analyzing the actual lab test results and/or patients answers for particular PPI questions. We speculate that the missing pattern may be related to the disease classification and disease subgroups, especially for the lab codes. However, we note that the missing pattern for the PPI data might be strongly confounded by other nonclinical characteristics such as age, gender and social determinants of health.

We used a 20-topic MixEHR to model the distinct distribution of these six data types. Our preliminary analysis showed that more than 20 topics produced degenerate models. First, we sought to quantitatively examine whether the patient topic mixtures provide useful information for accurate classification of bipolar disorder. To this end, we divided the 187 patients into five folds (Fig. 4a). We trained MixEHR on the four folds and then a logistic regression (LR) classifier that uses the patient topic mixture in the four folds to predict the bipolar disorder label. In this evaluation, we removed ICD-9 code category 296 from all patients as it codes for bipolar disorder. We then applied the trained MixEHR and LR to predict the BD labels of patients in the validation fold based on their MixEHR 20-topic mixture. As a comparison, we also applied two other unsupervised learning approaches, namely Latent Dirichlet Allocation (LDA) with 20 topics and Restricted Boltzmann Machine (RBM) with 20 hidden units.

**Longitudinal MIMIC-III EHR code prediction details**. Because our current model is not longitudinal, we developed a pipeline that combines MixEHR topics with recurrent neural network (RNN) with Gated Recurrent Unit (GRU) (Fig. 6a). We first trained MixEHR on the EHR data for ~39,000 patients with single admission in MIMIC-III. We then used the trained MixEHR to infer topic mixture at each admission for the 7541 patients with multiple admissions. Then we used as input the inferred topic mixture at the current admission (say at time $t$) to the RNN to auto-regressively predict the diagnostic codes at the next admission at time $t + 1$. Here, MixEHR uses all of the data types from MIMIC-III. Specifically, we used a 128-dense layer to embed the input topic mixture before passing it to the two 128-GRU layers followed by a dense layer connected to the CCS medical code output classes. We set the L2 weight penalty to 0.001 and dropout rate 0.5 based on preliminary results and existing literature.

As a comparison, we ran one of the recently developed state-of-the-art deep-learning frameworks called Graph-based Attention Model (GRAM)[32]. GRAM requires a knowledge graph such as the CCS multi-level diagnosis hierarchy or ICD-9 taxonomy and at least two admissions for each patient. Therefore, we only trained GRAM on the 7541 patients' admissions using only their ICD-9 codes. For MixEHR+RNN and GRAM, we randomly split the multi-admission patients into 80% for training and 20% for testing. Same as the evaluation performed by Choi et al., we prepared the true labels by grouping the ICD-9-CM codes into 283 groups using the Clinical Classification Software (CCS)[32].

In terms of evaluating prediction accuracy of specific medical codes, we found that all of the models including GRAM and Doctor AI do not perform well on the low-frequent medical codes (with median accuracies equal to 0 and mean accuracies below 25% accuracy range), and the accuracy estimates are unstable for the rare medical codes, depending on the training and testing split. Therefore, we

focused on evaluating the predictions of the 42 medical codes each observed in at least 1000 training admissions. We then calculated the accuracy at the top 20 predictions for each CCS diagnosis code. For each medical code, we counted the number of true positives among the top 20 predicted code for each admission and then divided it by the number of total positives.

**Quebec CHD EHR code prediction details**. In this database, each patient had on average 28 years medical follow-up history (1983–2010). The input data have two data types namely, ICD-9 or ICD-10 code and procedure intervention code. We designed an RNN that takes as input both the raw EHR data and the multimodal topic mixture (over the two ICD and procedure data types) inferred by MixEHR at each visit block $t$ and predicts as outputs the first three digits of the diagnostic ICD code at the next visit block $t + 1$ (Fig. 6c). Similar to the design from Doctor AI[27], we defined a visit block as a 1-month interval that contains at least one visit from the patient. For instance, suppose a patient visits multiple times in January and does not visit since then until May. We pool the data over those visits in January. We ignore the months where there is no visit by that patient (i.e., February to April). We predict diagnostic code for the next visit block $t + 1$ in the subsequent month that contains at least one visit by that patient (i.e., May in the above hypothetical example).

Here, we sought to assess whether there is additional information provided by MixEHR that is not captured by the fully connected dense layer of the RNN in terms of the code prediction improvement. For the ease of reference, we call this approach as MixEHR+RNN to distinguish it from the baseline RNN that takes only the raw EHR as input (i.e., Doctor AI). For the baseline RNN, we use 100-dimensional hidden embedding (i.e., fully connected dense layer) to encode the observed EHR code in the input. For the MixEHR+RNN model, we used 50-dimensional hidden embedding to encode the observed EHR code augmented with 50-topic MixEHR for the topic mixture embedding (Supplementary Fig. 21). Therefore, both models have 100-dimensional latent embeddings to encode the input features. The rest of the architecture stays the same for both RNN models. Details are described in Supplementary Note.

For evaluation, we randomly split the 80,000 patients into 80% of the patients for training and 20% of the patients for testing. For MixEHR+RNN, we first trained MixEHR on all of the visit blocks of the training patients. Then, we trained an RNN that takes the raw input concatenated with the MixEHR-inferred 50-topic mixture at each visit block of the training patients to predict the diagnostic code at the next visit block. For testing, we applied the trained MixEHR+RNN and the baseline RNN to predict the diagnostic code in the next visit block of the test patients based on all of the previous visit block(s). For MixEHR+RNN and the baseline RNN models, we recorded the prediction accuracy across all visit blocks (excluding the first visit block) for each patient in terms of AUROC and AUPRC for each diagnostic code.

**MIMIC-III lab results imputation details**. In total, there are 53,432 unique features/variables over six data types in the MIMIC-III. Among these, 564 variables are lab tests. We used all of the variables (notes, ICD, lab, prescriptions, DRG, CPT) across all of the six data types to learn the topic distribution from the training set and performed imputation of lab results on the testing set (Fig. 7a). For the observed lab tests of each patient, we used the FLAG column that indicates normal or abnormal level for the lab results. We counted for each admission the number of times the lab results were normal and abnormal. A patient at the same admission can exhibit both normal and abnormal states zero or more times for the same lab test. In this case, we recorded the frequency at each state for the same admission.

The goal of our analysis is to impute missing lab test results based on all of the other information of the patient's admission. Therefore, we did not distinguish whether the admissions came from the same patients and focused on imputing lab results within the same admission. Most patients only have one admission in the MIMIC-III dataset: among the 46,522 patients only 7541 patients (16%) have more than one admission. Therefore, the conclusion will likely remain the same if we were to aggregate information from multiple admissions for the same patient.

We randomly split the total 58,903 admissions into 80% for training and 20% for testing. For the training set, we trained our MixEHR to learn the topic distribution and the 50-topic mixture memberships for each training admission (Fig. 7a, step 1). We then iterated over the admissions in the test set. For every lab test observed in the test admission, we first masked its test result and imputed the test result as follows. We first inferred the 50-topic mixture memberships for each test admission using the topic distribution learned from the training set (Fig. 7a, step 2). We then found the 25 most similar admissions to the test admission from the training set based on the Euclidean distance of their topic mixture memberships (Fig. 7a, step 3). These 25 admissions must have the target lab test observed. We then took the average of the frequency over the 25 most similar training admissions as our imputed lab result for the target lab test in the test admission.

This procedure was repeated for every lab test in every test admission. We recorded the predicted lab results and the true lab results for evaluation purposes. The same way was used to evaluate the CF-RBM method. As a comparison, we trained conditional factored Restricted Boltzmann Machine (CF-RBM)[5] with 100 hidden units only on the lab tests. For MixEHR and CF-RBM, we used 80% of the admissions for training and 20% of the admissions for testing. Here we did not

distinguish whether the admissions came from the same patients but rather focused on imputing lab results within the same admission.

To further compare our proposed approach, we took a simpler imputation strategy by averaging. For each lab test, we took the average of the observed lab results over the training admissions. We then used the average frequency as the predictions for the lab results on the 20% testing admissions.

**MIMIC-III mortality prediction details**. We first trained MixEHR on the ~39 K patients with single-admissions to learn the disease topics. We used the trained MixEHR to infer topic mixture of each admission for the patients who had more than one admission. We took the last two admissions that are within 6 months apart, which gave us around 4000 multi-admission patients. We used the topic mixture inferred for the second-last admission as an input to a classifier to predict the mortality outcome at the discharge of the last admission (i.e., as indicated by the EXPIRED flag in the Discharge location of the last admission). To realistically evaluate our model, we filtered out Discharge Summary from the clinical notes in all of the data (under CATEGORY of the notes) because they usually contain conclusion about patients' critical conditions. We then performed a 5-fold cross-validation to evaluate the prediction accuracy. We experimented with $K = 50$ and 75 topics. As baseline models, we tested LDA on a flattened EHR matrix over all six data types, Principal Component Analysis (PCA) with 50 Principal Components (PCs), Sparsity-inducing Collected Non-negative Matrix Factorization (SiCNMF) on the same input data[19]. We ran SiCNMF with the default settings with lower rank set to 20. Lastly, we trained elastic net directly on the raw input data from the ~4000 patients' second-last admission to predict the morality in their last admissions upon discharge.

Moreover, we designed another experiment that used the earliest/first admission to predict the mortality in the last admission. We removed patients if their earliest admission lasted longer than 48 h based on the difference between the admission time and discharge time within the same admission. This also gave us 4040 patients whose first in-hospital stay was shorter than 2 days. Same as the above, we performed a fivefold CV on these patients as above using the pre-trained unsupervised models on the single-admission patients' data to generate topic mixtures on the first admissions as input to a supervised classifier (i.e., elastic net), which then predicts the mortality in the last admission upon discharge.

**Reporting summary**. Further information on research design is available in the Nature Research Reporting Summary linked to this article.

## Data availability

The MIMIC-III data analyzed in this paper are publicly available through PhysioNet (http://mimic.physionet.org). Mayo Clinic and Quebec CHD data are not publicly accessible due to restricted user agreement.

## Code availability

We implemented MixEHR in C++ as a standalone Unix command-line software using OpenMP for multi-threaded inference. It allows an arbitrary number of discrete data types and discrete states per EHR feature. The software is available as Supplementary Software 1 or at https://github.com/li-lab-mcgill/mixehr.

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

## Acknowledgements

Y.L. is supported by Natural Sciences and Engineering Research Council (NSERC) Discovery Grant (RGPIN-2019-0621), Fonds de recherche Nature et technologies (FRQNT) New Career (NC-268592), and Canada First Research Excellence Fund Healthy Brains for Healthy Life (HBHL) initiative New Investigator award (G249591). The funding for the Mayo Clinic bipolar disorder study is provided by Marriott Foundation. Mayo Clinic Center is funded by Individualized Medicine. The Québec CHD Database is funded by Dr. Marelli's Canadian Institute of Health Research Foundation Grant Award #35223. We thank Mathieu Blanchette for the helpful comments on the paper.

## Author contributions

Y.L. and M.K. conceived the initial idea. Y.L. developed and implemented the method, processed and analyzed the data, and wrote the initial draft of the paper. P.N. ran the experiments on the EHR code prediction and mortality prediction using MIMIC-III data. Y.W. and A.A.K.D. helped running part of the lab imputation experiments. T.O., J.M.B., J.E.O., and M.A.F. provided the Mayo Clinic dataset. E.R. performed the data extraction and de-identification of the EHR data for both the bipolar and the control cohorts. W.L., Y.M., and Z.W. helped running the Mayo Clinic part of the experiment. T.O., J.M.B., and E.R. helped analyze the results. A.M. provided the data from Quebec CHD Database. A.L. and L.G. processed the data. X.H.L. implemented the MixEHR+RNN model and ran the experiments on the Quebec CHD Database with the help from Z.W. A.L. and A.M. helped interpreting the results from the Quebec CHD Database. Y.A. and J.D. performed the initial data processing and helped interpreted the initial results. M.K. edited the initial draft of the paper. Y.L. and M.K. supervised the project. All of the authors contributed to the final writing of the paper.

## Competing interests

The authors declare no competing interests.
