## [Peer Review File · Nature Communications]

Reviewers' comments:

Reviewer #1 (Remarks to the Author):

This paper proposes a probabilistic topic model to learn latent disease topics from electronic health records. This study is interesting, however, this reviewer has major concerns regarding the dataset and methods.

Major concerns:

1. Dataset.

1.1. MIMIC III data contain only critical care data without patients' longitudinal dataset. This reviewer concerns about using MIMIC data in this case.

1.2. Demographics of the dataset is not provided. This reviewer understands MIMIC data may not release such information. But understanding the subjects would be crucial for making any clinical/medical statement.

1.3. Is there a control group to verify the statement regarding topics correlated with age/psychiatric traits?

2. Method. The proposed method MixEHR is a novel probability model to learn latent disease topics from EHR. However, a major concern about methods is the lack of comparison with SOTA methods. For example, neural networks have been used to learn patient and disease representations from EHRs. Examples include: [1] Deep patient: an unsupervised representation to predict the future of patients from the electronic health records. [2] Doctor ai: Predicting clinical events via recurrent neural networks. [3] GRAM: graph-based attention model for healthcare representation learning. How are these methods compared with the proposed methods. Even for LDA, there are more advanced topic models. But this study does not show the advantage of MixEHR over these models.

3. Experiments are limited to MIMIC data. The generalizability of the proposed method would be a significant issue, but not addressed in this study.

Reviewer #2 (Remarks to the Author):

The study proposes MixEHR, a multi-view Bayesian framework, for EHR data integration and modeling. The analogy idea is simple - each patient as a "document" and each disease phenotype as a "topic" - and based on latent topic modeling. The math is neat and joint distribution of data observations is helpful. And the experimental results on MIMIC-iii data are presented.

1. The major problem of the study is MixEHR completely ignores the longitudinal aspect of the EHR data, even without any concept of index date or time window of EHR data analysis. Since EHR data usually spans decades, it is not convincing that a patient's disease phenotype is the same across multiple years. Some experiments on temporal trajectory are beneficial - even via sliding windows.

2. Another major problem is the conducted experiments lack clinical meaning. e.g., the EHR code prediction task is to predict whether a target code exists when there are other codes and lab tests, etc. However, what clinicians expect is a clinical decision support tool to predict all EHR codes based on their observations (e.g., medical histories, lab data). Such an imputation setting is not practical. Similarly, the mortality risk prediction task is not well-described - how to split the training/testing? How to choose those positive/negative samples? How to deploy the mortality risk prediction to clinical settings? All need some additional design and experiments.

3. Although the paper includes some literature review, it doesn't include any necessary performance comparisons to existing algorithms - is MixEHR better than state-of-the-arts? For example:

- For code prediction and mortality prediction tasks, performance comparisons to some of the

references 32-37 are necessary.

- For disease phenotyping/grouping tasks, performance comparisons to some of the references 29, 30, 41, 48 are necessary.
- For EHR/lab imputation tasks, there are also some related works other than MICE. Please review and compare in the experiments.

4. While the paper is well-written in grammar, it contains too many contents and not well-structured. Section 4.1 - MIMIC data processing contains little information.

Point-by-Point Response to Reviewers

Overview

We would like to thank both Reviewers for the thoughtful and constructive comments. We have made substantial amounts of effort to address every single comment.

To briefly summarize,

1. **Longitudinal:** We performed longitudinal analysis on the MIMIC-III data in terms of predicting diagnostic code in future admissions. Furthermore, we added another separate outpatient dataset extracted from the Quebec Congenital Heart Disease (CHD) Database with 80,000 patients and 28-year of medical history follow-up;
2. **Modeling:** We presented a new deep learning architecture by incorporating our MixEHR topic mixture as embedding into a recurrent neural network to improve longitudinal code prediction over the existing state-of-the-art methods (SOTA);
3. **Improved clarity and experiments:** We have improved our experimental design on the MIMIC-III data in terms of code prediction, lab imputation, and mortality prediction;
4. **Method comparison:** We added comparison with the SOTA methods and demonstrated competitive performance by our proposed MixEHR model;
5. **Generalizability:** In addition to the MIMIC-III dataset, we demonstrated our model on two more datasets namely Bipolar Disorder patient EHR data from Mayo Clinic and Quebec CHD Database. In each application, we compared our method with the SOTA methods on EHR code prediction and phenotype prediction, respectively.

The revised contents are highlighted in **Dark Red** in the main text.

Reviewer #1 (Remarks to the Author):

This paper proposes a probabilistic topic model to learn latent disease topics from electronic health records. This study is interesting, however, this reviewer has major concerns regarding the dataset and methods.

Major concerns:

1. Dataset.

1.1. MIMIC III data contain only critical care data without patients' longitudinal dataset. This reviewer concerns about using MIMIC data in this case.

Our Response: Indeed, a majority of the patients in MIMIC-III only had one admission record. On the other hand, there are ~7500 patients in MIMIC-III who had at least two admissions. This precious dataset has been used in several existing papers to demonstrate longitudinal modeling. Choi et al. (PMLH 2016) applied Doctor AI, which is a recurrent neural network (RNN) with gated recurrent units (GRU), to the MIMIC dataset limiting to the 2695 patients who made at least two visits. Choi et al. (KDD 2017) applied GRAM (a graph-based attention network combined with RNN) to ~7500 patients from MIMIC-III to model the temporal admissions among those patients. Rajkomar et al. (Nature Digital Medicine, 2018) also applied an RNN to MIMIC-III dataset.

We strongly agree with the reviewer on the importance of modeling longitudinal EHR dataset. In this revision, we have included longitudinal analysis of the MIMIC-III data using the ~7500 patients with at least two admissions. We focused our analysis on the longitudinal EHR code prediction in MIMIC-III data (**Section 3.4.2**). We also made comparisons with Doctor AI (Choi et al., PMLH 2016) and GRAM (Choi et al., KDD 2017) in terms of the prediction accuracy. Because our current MixEHR is not a longitudinal model, we present a novel framework that feeds the MixEHR topic mixture for the current admission t for each patient as input to the GRU in a RNN model to make prediction of diagnostic code at the next admission $t+1$ (**Figure R3a**). We demonstrated competitive performance with our MixEHR+RNN approach (**Figure R3b**). Please see details from our response to your **Comment 2** below.

We are fully aware of the limitation of our analysis because of the small sample size and short medical history of these 7500 MIMIC-III patients (on average fewer than 3 years of follow-up per patient). Deep learning

approaches typically require a large number of training examples that are often hundreds of thousand times bigger than the feature dimension. Also, an RNN is able to learn a more accurate estimate of the patient state as it sees **a long history of patient records**. Although GRAM is able to operate on an existing knowledge graph to remedy the problem, it is ideal to apply GRAM to a much larger dataset (e.g., over 250,000 patients as demonstrated by Choi et al). Unfortunately, access to these datasets are often restricted due to the local hospital policy that protects patient privacy and government regulation. We made clear comments on this limitation in **Section 3.4.2** and **Discussion Section 4**.

Fortunately, we did manage to establish a collaboration with Dr. Ariane Marelli's team from McGill Adult Unit for Congenital Heart Disease Excellence (MAUDE Unit). We demonstrated our model on 80,000 patients with on average 28 years of medical history follow-up (**Section 3.5**). We used a similar MixEHR+RNN framework as the above experiment but with some architecture changes (**Figure 6c**; **Supplementary Figure S21**). Here we predict future diagnostic code at the next visit block based on the previous visit blocks. The visit block is defined as the month, in which the patient visited the hospital at least once. Compared to the baseline RNN, we observed a significant improvement when augmenting the RNN with MixEHR-inferred topic mixture at each visit block. We are careful in designing this experiment so that no information for the test patients were used to train MixEHR or RNN. Therefore, the prediction improvement on the next visit block is solely attributable to the extra information distilled from our MixEHR model on the previous visit blocks of the test patient. Please see details in **Section 3.5** and our response to your **Comment 3** on the generalizability of our model below.

1.2. Demographics of the dataset is not provided. This reviewer understands MIMIC data may not release such information. But understanding the subjects would be crucial for making any clinical/medical statement.

Our response: Thank you for the suggestion. We have included demographic information in our analysis. In this revision, we added the comparison of in-admission mortality prediction. We compared the performance of predicting mortality within the same admission, which is a much easier task than the one presented in the main text. We split the admissions into 80% for training and 20% for testing. We then trained MixEHR model on the training set and a logistic regression (LR) classifier that used the input topic mixture and/or demographic information including sex, age, religion, ethnicity, and marital status of the admitted patient and predicted mortality labels in that admission. We then applied the trained MixEHR and LR on the testing set of the admissions. The performance was evaluated based on ROC and precision-recall curves. Our results show little benefits of adding demographic information in the mortality prediction (**Figure R1**). However, we acknowledge its importance and will continue exploring this in our future work.

We added the above analysis in **Section 3.6 Mortality risk prediction in MIMIC-III** (page 12 in the main text).

Figure R1. Mortality prediction using demographic information using the MIMIC-III dataset. We compared the performance of predicting mortality within the same admission, which is a much easier task than the one presented in the main text. We split the admissions into 80% for training and 20% for testing. We then trained MixEHR model on the training set and a logistic regression (LR) classifier that used the input topic mixture and/or demographic information including sex, age, and religion of the admitted patient and predict mortality label in that admission. We then applied the trained MixEHR and LR on the testing set. The performance was evaluated based on ROC and precision-recall curves.

1.3. Is there a control group to verify the statement regarding topics correlated with age/psychiatric traits?

Our response: Thank you for the suggestion. If we understood the question correctly, we will need to use the topics we learned from MIMIC-III data as a guide to score patients from another independent dataset. We unfortunately do not have these datasets to support the robustness of the age-related and psychiatry-related topics. We are completely open to further suggestions to improve this aspect of the analysis. On the other hand, regarding model's ability to learn psychiatric topics, we did test our model on a separate dataset on 187 subjects collected from Mayo Clinic (**Section 3.3**). Please see our response to your **Comment 3** below.

2. Method. The proposed method MixEHR is a novel probability model to learn latent disease topics from EHR. However, a major concern about methods is the lack of comparison with SOTA methods. For example, neural networks have been used to learn patient and disease representations from EHRs. Examples include: [1]Deep patient: an unsupervised representation to predict the future of patients from the electronic health records. [2] Doctor ai: Predicting clinical events via recurrent neural networks. [3] GRAM: graph-based attention model for healthcare representation learning. How are these methods compared with the proposed methods. Even for LDA, there are more advanced topic models. But this study does not show the advantage of MixEHR over these models.

Our response: Thank you for the suggestion. In this revision, we performed a rigorous comparison with the current SOTA methods namely Deep Patient (Miotto et al., Scientific Report 2016), Doctor AI (Choi et al., arXiv 2016), and GRAM (Choi et al., KDD 2017).

Comparison with Deep patient:

Because both our proposed MixEHR and Deep Patient (DP) are unsupervised, we are comparing how the low-rank embedding information learned by each method can be used in a prediction task. To this end, we evaluated the performance of predicting mortality within the same admission, which is a much easier task than predicting future mortality presented in the main text **Section 3.7**. We randomly split all of the 58903 admissions from MIMIC-III into 80% for training and 20% for testing. We then trained MixEHR model on the training set using all of the data types (notes, ICD-9, CPT, DRG, lab, prescription) and a logistic regression (LR) classifier that used the input topic mixture to predict mortality label in that admission.

In our comparison with Deep Patient, since there was no published code available, we implemented the stacked denoising autoencoder to the best of our abilities, closely following the steps mentioned in the paper by the authors. We did not follow the elaborate data preprocessing pipeline used in the paper as our dataset (MIMIC-III) is different from the one used by Miotto et al. (Sci Rep. 2016). Our implementation had 3 layers of autoencoders with 500 hidden units (as in the original paper) and coded using Keras library. We used the Deep Patient representation as input to a classifier (i.e., logistic regression) to predict mortality and compared the accuracy with MixEHR in terms of area under the precision-recall (AUPRC) and area under the ROC (AUROC) (**Figure R2**). As we can see, the performance of DP is not good. This is perhaps due to the small sample size and the lack of carefully designed preprocessing pipeline as Miotto et al. demonstrated on their own EHR data. Notably, in their original paper, DP was applied to a dataset containing 700,000 patients as opposed to only 58,903 admissions or only 46,501 distinct patients.

We described this in our revision at the end of **Section 3.6 Mortality risk prediction in MIMIC-III** and the implementations were described in **Section 5.9 Implementations of existing deep learning methods**.

Comparison with Doctor AI and GRAM:

For experiments comparing our method with GRAM (Choi et al., KDD 2017) and DoctorAI (Choi et al., PMLH 2016), we used the open source implementation made available by the authors of the papers found at <https://github.com/mp2893/gram> and <https://github.com/mp2893/doctorai>, respectively.

In the MIMIC-III data, there are 7541 patients who were admitted to the hospital more than once. This allows us to evaluate how well we can predict the diagnostic code in the next admission based on the information from the previous admission(s) for the same patients.

Because our current model is not longitudinal, we developed a novel pipeline that combines MixEHR topic mixtures with recurrent neural network (RNN) with Gated Recurrent Unit (GRU) (**Figure R3a**). We first trained MixEHR on the EHR data for 39,000 patients with single-admission in MIMIC-III. We then used the trained MixEHR to infer topic mixture at each admission for the 7541 patients with multiple admissions. Then, we used as input the inferred topic mixture at the current admission (say at time t) to the RNN to autoregressively predict the diagnostic codes at the next admission at time $t+1$. Here MixEHR uses all of the data types from MIMIC-III.

Because GRAM requires a knowledge graph such as the ICD-9 taxonomy and at least two admissions for each patient, we only trained GRAM on the 7541 patients' admissions using only their ICD-9 codes. For MixEHR+RNN and GRAM, we randomly split the multi-admission patients into 80% for training and 20% for testing. Same as the evaluation performed by Choi et al, we prepared the true labels by grouping the ICD9-CM codes into 283 groups using the Clinical Classification Software (CCS). We then calculated the accuracy at the top 20 predictions for each CCS diagnosis code, i.e., at each admission, we counted the number of times the target diagnosis codes were among the top 20 predicted code. We binned accuracy by five percentile (0-20, 20-40, 40-60, 60-80, 80-100) based on the frequency the EHR code is observed among the patients. The rationale behind this is that the lower the frequency the code observed the more difficult it is to accurately predict the code. By leveraging the correlation structure information between rarely observed codes and commonly observed codes via the latent topics, we hypothesize that our approach can achieve comparable prediction accuracy as GRAM without relying on the existing ICD-9 knowledge graph. Indeed, we observe quite a competitive performance of our approach compared to GRAM (**Figure R3b**). We also compared with Doctor AI (Choi et al., PMLH 2016), which is another RNN framework, using the same dataset and the same metric. The performance of Doctor AI is worse than GRAM and MixEHR+RNN (**Figure R3b**).

Deep learning approaches typically require a large number of training examples that are often hundreds of thousand times bigger than the feature dimension. Also, an RNN is able to learn a more accurate estimate of the patient state as it sees a long history of patient records. This is not the case in our experiment because the number of patients or number of admissions is much smaller than the total number of features. Also, MIMIC-III stores less than 3 years of medical history for most patients who have more than one admission. Although GRAM is able to operate on an existing knowledge graph to remedy the problem, it is ideal to apply GRAM to a much larger dataset. For example, Choi et al (PMLH 2016; KDD 2017) trained GRAM and Doctor AI models on over 250,000 patients. These datasets are not available to us. We also note that we may also improve the performance of GRAM by modifying the code to do an unsupervised pre-training on the ~39,000 single-admission patients before the supervised training on the multi-admission patients. This goes beyond the scope of this study.

We presented our perspective on the deep learning in **Section 2 Related Work** and the above results in **Section 3.4.2 Longitudinal EHR code prediction in MIMIC-III data** in the revision. The implementations were described in **Section 5.9 Implementations of existing deep learning methods**.

Figure R2. Mortality prediction comparison between MixEHR and deep patient (DP) embeddings using the MIMIC-III dataset. We evaluated the performance of predicting mortality within the same admission, which is a much easier task than predicting future mortality presented in the main text. We randomly split the admissions into 80% for training and 20% for testing. We then trained MixEHR model on the training set and a logistic regression (LR) classifier that used the input topic mixture to predict mortality label in that admission. We also trained DP on the same training set and a logistic regression (LR) classifier that used the input DP embedding to predict mortality label in that admission. We then applied the trained MixEHR+LR and DP+LR on the testing set. The performance was evaluated based on ROC and precision-recall curves.

3. Experiments are limited to MIMIC data. The generalizability of the proposed method would be a

significant issue, but not addressed in this study.

Our response: In this revision, besides improving the experiments on the MIMIC data (Application 1), we have experimented our method on two additional datasets obtained through our collaboration with Mayo Clinic, Rochester, Minnesota, USA and McGill Adult Unit for Congenital Heart Disease Excellence (MAUDE Unit), Montreal, Quebec, Canada. We present the results of applying MixEHR on these two datasets in detail below.

Application 2: on Mayo Clinic Dataset:

To further demonstrate the utility of our approach in discovering meaningful multi-modal topics, we applied MixEHR to a separate dataset containing 187 patients including 93 bipolar disorder cases and 94 age- and sex-matched controls from Mayo Clinic (see **Section 5.10.2** for more details). Despite the small sample size, the patients are deeply phenotyped: there are in total 7731 heterogeneous EHR features across 5 different data categories including ICD-9 codes, procedure codes, patient provided information (PPI), lab tests, and prescription codes. In total, there are 108,390 observations.

Among 5 data categories, lab tests and PPI have two separate data features unlike the other 3 binary data categories. The lab and PPI data categories consist of (1) whether the data exist in EHR (e.g., whether a lab test was done or whether a patient answered a particular PPI question) and (2) actual data (e.g., test results for lab tests and patient's answer for each PPI question). Given the small sample size, we chose to model the first data feature (whether the data exist in EHR) to mimic the data structure used in the rest of data categories, recognizing that we may miss important information by not analyzing the actual lab test results and/or patient's answer for a particular PPI question. We speculate that the missing pattern may be related to the disease classification and disease subgroups, especially for the lab codes. However, we note that the missing pattern for the PPI data might be strongly confounded by other non-clinical characteristics such as age, gender and social determinants of health.

We used a 20-topic MixEHR to model the distinct distribution of these 6 data types. Our preliminary analysis showed that more than 20 topics produced degenerate models. First, we sought to quantitatively examine whether the patient topic mixtures provide useful information for accurate classification of bipolar disorder. To this end, we divided the 187 patients into five folds (**Fig. R4a**). We trained MixEHR on the four folds and then a logistic regression (LR) classifier that uses the patient topic mixture in the four folds to predict the bipolar disorder label. In this evaluation, we removed ICD-9 code category 296 from all patients as it codes for bipolar disorder. We then applied the trained MixEHR and LR to predict the BD labels of patients in the validation fold based on their MixEHR 20-topic mixture. As a comparison, we also applied two other unsupervised learning approaches, namely Latent Dirichlet Allocation (LDA) with 20 topics and Restricted Boltzmann Machine (RBM) with 20 hidden units. We observed superior performance of MixEHR+LR compared to LDA+LR and RBM+LR in both area under the ROC and area under the precision-recall curves (**Fig. R4b**).

We then trained MixEHR and LR on the full patient data and quantified the predictive information of the 20 topics based on the linear coefficients. We observed that topics M19 and M7 have the highest positive coefficients for bipolar disorder label (**Fig. R4c**). To confirm the finding, we visualized the 187 patient topic mixtures side-by-side with their BD diagnosis ICD-9 296 code (**Fig. R4d; Supplementary Data SD3**). Indeed, we find that M19 and M7 are highly correlated with the BD diagnosis label, and M19 and M7 may represent two distinct subgroups of BD patients.

We further investigated the differences between the M19 and M7 topics in terms of the underlying 7731 EHR codes. For each EHR code, we grouped the patients based on the presence and absence of that code. We then tested whether the two groups are significantly different in terms of their topic mixture membership under M19 or M7 using Wilcoxon Signed Rank one-sided tests. For the ease of interpretability, here we tested whether the patients with the code exhibit significantly higher topic mixture for M19 and M7, to assess positive enrichment for the topic. We note that these phenome-wide association studies (PheWAS) results were not corrected for multiple testing and thus we used the results as an exploratory purpose only.

While a large proportion of the significant EHR codes is associated with both the M7 and M19 topics, we also find interesting codes that are unique to each topic (**Fig. R5**). For instance, ICD-9 codes for suicidal ideation (V62.84), family history of psychiatric condition (V17.0), and bipolar type I disorder with the most recent severe depressed episode without psychotic behavior (296.53) are significantly associated with M19 but not M7, although both topics share 296.80 (Bipolar disorder, unspecified) and 296.52 (Bipolar I disorder, most recent episode (or current) depressed, moderate) codes for BD (**Fig. R5**). Certain lab tests (e.g., SLC6A4, HTR2A,

and cytochrome P450 enzymes) are also associated with M19 but not M7. Lithium lab test is also associated with high M19 mixture probabilities, but not M7. Thus it is possible that M19 patients may have had more severe symptoms and needed an increased use of pharmacogenomically-guided treatment. PheWAS for the PPI questions are displayed in **Supplementary Fig. S31**. Interpretation on the PPI PheWAS must be taken with caution due to caveats mentioned above. With this data, we are able to demonstrate a potential utility of our MixEHR approach to classify BD patients into potentially clinically meaningful categories that requires further investigation in larger dataset.

We presented this in **Section 3.3 Classification of bipolar disorder patients (Mayo Clinic data)** in the revision.

Application 3: 28-year longitudinal Quebec CHD Database:

We also evaluated the code prediction accuracy on the longitudinal outpatient data from the Quebec CHD database. In this database, each patient had on average 28 years medical follow-up history (1983-2010). The input data have two data types namely, ICD-9 or ICD-10 code and procedure intervention code. We designed an RNN that takes as input both the raw EHR data and the multi-modal topic mixture (over the two ICD and procedure data types) inferred by MixEHR at a visit block t and predicts as outputs the first 3 digits of the diagnostic ICD code at the next visit block $t + 1$ (**Figure R3c**).

Similar to the design from Doctor AI (Choi PMLH 2019), we defined a visit block as a one-month interval that contains at least one visit from the patient. For instance, suppose a patient visits multiple times in January and does not visit since then until May. We pool the data over those visits in January. We ignore the months where there is no visit by that patient (i.e., February to April). We predict diagnostic code for the next visit block $t + 1$ in the subsequent month that contains at least one visit by that patient (i.e., May in the above hypothetical example).

Here we sought to assess whether there is additional information provided by MixEHR that is not captured by the fully connected dense layer of the RNN in terms of the code prediction improvement. For the ease of reference, we call this approach as MixEHR+RNN to distinguish the baseline RNN that takes only the raw EHR as input (i.e., Doctor AI). For the baseline RNN, we use 100-dimensional hidden embedding (i.e., fully connected dense layer) to encode the observed EHR code in the input. For the MixEHR+RNN model, we used 50-dimensional hidden embedding to encode the observed EHR code augmented with 50-topic MixEHR for the topic mixture embedding (**Supplementary Fig. S21**). Therefore, both models have 100-dimensional latent embeddings to encode the input features. The rest of the architecture stays the same for both RNN models. Details are described in **Supplementary Information Section S5**.

For evaluation, we randomly split 80,000 patients into 80% of the patients for training and 20% of the patients for testing. For MixEHR+RNN, we first trained MixEHR on all of the visit blocks of the training patients. Then, we trained an RNN that takes the raw input (and the MixEHR-inferred 50-topic mixture) at each visit block of the training patients to predict the diagnostic code at the next visit block. For testing, we applied the trained MixEHR+RNN and baseline RNN to predict the diagnostic code in the next visit block of the test patients based on all of the previous visit block(s). For MixEHR+RNN and baseline RNN models, we recorded the prediction accuracy across all visit blocks (excluding the first visit block) for each patient in terms of AUROC and AUPRC for each diagnostic code.

We observed a significant improvement in terms of both AUPRC (Wilcoxon Signed Rank Tests p -value < 0.0408) and AUROC (Wilcoxon Signed Rank Tests p -value $< 5.35e-55$) (**Figure R3d**). Therefore, adding MixEHR significantly improves the EHR code prediction over the baseline RNN model. We also observed that the learned weights connected to the 50-topic mixture exhibit higher magnitude than the learned weights connected to the concatenated dense layer embedding (**Supplementary Fig. S19**). This means that the network relies heavily on the topic mixture to make accurate predictions. Because our dataset is focused on the CHD patients, we checked the prediction accuracy on ICD-9 code 428. Both models achieve 93% AUROC and 33% AUPRC for predicting 428 ICD-9 code (**Supplementary Data SD4**).

Compared to the baseline RNN, our MixEHR+RNN provided additional interpretation thanks to the topic model component. In particular, we discovered two interesting topics namely M43 and M1 that are highly related to heart failure ICD-9 code 428 (**Supplementary Fig. S20**). Specifically, M43 involves not only ICD-9 code 428.9 for heart failure but also code 518.4 for acute lung edema, code 290.9 for senile psychotic condition, code

428.0 for congenital heart failure, code 402.9 for hypertensive heart disease without HF, and code 782.3 for edema. Indeed, edema is known to be the precursor for heart failure among many patients. Interestingly, topic M1 characterizes a different set of heart-related diseases such as rheumatic aortic stenosis (code 395.0), secondary cardiomyopathy (code 425.9), and left heart failure (code 428.1).

Procedural codes 1-71, 1-9115, 1-9112 all indicate billing for more complex care either by a family doctor or specialists in patients who have impaired mobility, prolonged hospitalization beyond 15 days and/or admission to short-stay units for patients unable to go home, typically requiring diuretics for pulmonary edema. Therefore, most of the top EHR codes under topic M43 are clinically relevant predictors of heart-failure events in clinical practice.

Notably, our data-driven model is not protected from identifying confounder events such as 'senile psychotic condition', which does not have direct biological connection with heart-failure. We interpret this to mean that high thresholds of predictive probability should be applied to models that are purely data-driven in order to ensure optimal specificity. Additional parameters such as the inclusion of measures of time between topics that represent events could be helpful in excluding confounder events.

In summary, our data-driven approach, using a large number of heterogeneous topics over long observation periods, has the potential of leading to clinically meaningful algorithms that will help clinicians anticipate high disease burden events in clinical practice. Our findings can inform the growing emphasis on proactive rather than reactive health management as we move into an era of increasing precision medicine.

We presented these results in **Section 3.5** in the revision.

a. Workflow for evaluating bipolar disorder classification

MixEHR unsupervised learning

Supervised learning

Predicting held-out patient

b. Evaluation of classification accuracy

c. Linear coefficient of the 20 predictive topics

d. Inferred 20-topic mixture of 187 patients

Figure R4. Classification of bipolar disorder patients using the Mayo Clinic dataset. **a.** Workflow to evaluate classification accuracy of bipolar disorder in a 5-fold cross validation (CV). **b.** Classification accuracy of 5-fold CV in ROC and precision-recall curves. **c.** Linear coefficients of the 20 topics ranked by decreasing order. **d.** Inferred 20-topic mixture of 187 patients. The patients with and without bipolar disorder diagnosed were colored in red and green respectively on the left side of the dendrogram.

Figure R5. Phenome-wide association studies of the 6285 EHR codes on the two Bipolar Disorder mixture topics (M7 and M19) based on the Mayo Clinic dataset. For each EHR code, we tested the difference between patient groups with and without that code in terms of M19 and M7 topic membership probabilities. The labelled EHR codes are the ones with Wilcoxon Signed Rank one-sided test p-values < 0.001. The red, blue, and black color indicate the codes that are significant in only M19, in only M7, and in both M19 and M7, respectively. **Please see Figure 5 in the main text for a higher resolution image of this figure.**

Reviewer #2 (Remarks to the Author):

The study proposes MixEHR, a multi-view Bayesian framework, for EHR data integration and modeling. The analogy idea is simple - each patient as a "document" and each disease phenotype as a "topic" - and based on latent topic modeling. The math is neat and joint distribution of data observations is helpful. And the experimental results on MIMIC-iii data are presented.

1. The major problem of the study is MixEHR completely ignores the longitudinal aspect of the EHR data, even without any concept of index date or time window of EHR data analysis. Since EHR data usually spans decades, it is not convincing that a patient's disease phenotype is the same across multiple years. Some experiments on temporal trajectory are beneficial - even via sliding windows.

Our response: Thank you for the suggestion. We strongly agree with the reviewer on the importance of modeling longitudinal EHR dataset. In this revision, we included the longitudinal analysis of the EHR data using the ~7500 patients that have at least two admission records in the MIMIC-III dataset. We focused our analysis on the longitudinal EHR code prediction in MIMIC-III data (**Section 3.4.2**). We also made comparisons with Doctor AI (Choi et al., PMLH 2016) and GRAM (Choi et al., KDD 2017) in terms of the code prediction accuracy in the next admission using earlier admissions of the same patient. Because our current MixEHR is not a longitudinal model, we present a novel framework that feeds the MixEHR topic mixture for the current

admission t for each patient as input to an RNN model to make prediction of diagnostic code at the next admission $t+1$ (**Figure R3a and Figure 6a in the main text**). We demonstrated competitive performance with our MixEHR+RNN approach (**Figure R3b and Figure 6b in the main text**).

Through collaboration with Dr. Ariane Marelli's team from McGill University, we demonstrated our MixEHR by modeling the longitudinal EHR code for 80,000 patients with 28 year medical history follow-up (**Section 3.5**). We used a similar MixEHR+RNN framework as the above experiment but with some architecture changes (**Figure R3c or Figure 6c in the main text**). Here we predicted the future diagnostic code at the next visit block in one month interval based on the patient's earlier visit blocks. Compared to the baseline RNN, we observed a significant improvement when augmenting the RNN with MixEHR-inferred topic mixture at each visit block. We were very careful in designing this experiment so that no information for the test patients were used to train MixEHR or RNN. Therefore, the prediction improvement on the future admission is solely attributable to the extra information distilled from our MixEHR model on the earlier visit blocks of the test patient. Please see details in **Section 3.5** and our response to your **Comment 3** on **code prediction**.

2. Another major problem is the conducted experiments lack clinical meaning. e.g., the EHR code prediction task is to predict whether a target code exists when there are other codes and lab tests, etc. However, what clinicians expect is a clinical decision support tool to predict all EHR codes based on their observations (e.g., medical histories, lab data). Such an imputation setting is not practical. Similarly, the mortality risk prediction task is not well-described - how to split the training/testing? How to choose those positive/negative samples? How to deploy the mortality risk prediction to clinical settings? All need some additional design and experiments.

Our response: These are very useful points. In this revision, we made a clear distinction between the retrospective code prediction (**Section 3.4.1**) and the future code prediction (**Section 3.4.2** and **Section 3.5**). The former has its value in diagnosing the code entered in the existing EHR records and making suggestions about the potentially incorrectly entered code. This is in line with the applications in collaborative filtering for recommender systems such as imputing Netflix moving ratings. In our context, this is recommending potentially missing or incorrectly entered code based on the expectation of the EHR code.

As this reviewer kindly pointed out, however, it is more practical to predict EHR code in the future admission for a new test patient by modeling the clinical histories of the patient. We have done this experiment in **Section 3.4.2** and **Section 3.5** in comparison with the state-of-the-art (SOTA) methods. Please see our response to your **Comment 3** on **code prediction** below for more details.

For mortality risk prediction, we have improved the experimental design and re-done the method comparison with additional SOTA methods (**Section 3.6**). Please see our response to your **Comment 3** on **mortality prediction tasks** below for details.

3. Although the paper includes some literature review, it doesn't include any necessary performance comparisons to existing algorithms - is MixEHR better than state-of-the-arts? For example:

- For **code prediction** and **mortality prediction** tasks, performance comparisons to some of the references 32-37 are necessary.

Our response: We added method comparison on the code prediction and mortality prediction by comparing the representative methods in the references 32-37. In code prediction, we compared Doctor AI (Choi et al., PMLH 2016) and GRAM (a graph-based learning combined with RNN model; Choi et al., KDD 2017). In mortality prediction, we compared with Deep Patient and SiCNMF. For the ease of discussion, we place the mortality risk prediction as part of the response to the comment on **disease phenotyping/grouping tasks** and focus on the **code prediction** here. We have done two sets of experiments, one on MIMIC-III data and one on the newly acquired Quebec CHD dataset.

Application 1 on MIMIC-III data:

In the MIMIC-III data, there are 7,541 patients who were admitted to the hospital more than once. This allows us to evaluate how well we can predict the diagnostic code in the next admission based on the information from the previous admission(s) for the same patients. Notably, this is complementary to the retrospective code prediction task (**Section 3.4.1**) and can provide further support to the clinical decision making.

Because our current model is not longitudinal, we developed a novel pipeline that combines MixEHR topics with recurrent neural network (RNN) with Gated Recurrent Unit (GRU) (**Figure R3a**). We first trained MixEHR on the EHR data for 39,000 patients with single-admission in MIMIC-III. We then use the trained MixEHR to infer topic mixture at each admission for the 7541 patients with multiple admissions. Then we used as input the inferred topic mixture at the current admission (say time t) to the RNN to autoregressively predict the diagnostic codes at the next admission at time $t+1$. Here MixEHR uses all of the data types from MIMIC-III.

As a comparison, we ran one of the recently developed state-of-the-art deep learning frameworks called Graph-based Attention Model (GRAM) (Choi et al., KDD 2017) (**Section 5.9**). Because GRAM requires a knowledge graph such as the ICD-9 taxonomy and at least two admissions for each patient, we only trained GRAM on the 7541 patients' admissions using only their ICD-9 codes. For MixEHR+RNN and GRAM, we randomly split the multi-admission patients into 80% for training and 20% for testing. Same as the evaluation performed by Choi et al, we prepared the true labels by grouping the ICD9-CM codes into 283 groups using the Clinical Classification Software (CCS).

We then calculated the accuracy at the top 20 predictions for each CCS diagnosis code, i.e., at each admission, we counted the number of times the target diagnosis codes were among the top 20 predicted code and 0 otherwise. We binned accuracy by five percentile (0-20, 20-40, 40-60, 60-80, 80-100) based on the frequency the EHR code is observed among the patients. The rationale behind this is that the lower the frequency the code observed the more difficult it is to accurately predict the code. By leveraging the correlation structure information between rarely observed codes and commonly observed codes via the latent topics, we hypothesize that our approach can achieve comparable prediction accuracy as GRAM without relying on the existing ICD-9 knowledge graph. Indeed, we observe quite a competitive performance of our approach compared to GRAM (**Figure R3b**).

We also compared with Doctor AI (Choi et al., PMLH 2016) , which is another RNN framework, using the same dataset and the same metric (**Main text Section 5.9**). As expected, the performance of Doctor AI is worse than GRAM and MixEHR+RNN because of the small sample size and short medical history of the MIMIC-III data (**Figure R3b**). Deep learning approaches typically require a large number of training examples that are often hundreds of thousand times bigger than the feature dimension. Also, an RNN is able to learn a more accurate estimate of the patient state as it sees a long history of patient records. This is not the case in our experiment because the number of patients or number of admissions is much smaller than the total number of features. Also, MIMIC-III stores less than 3 years of medical history for most patients who have more than one admission.

Although GRAM is able to operate on an existing knowledge graph to remedy the problem, it is ideal to apply GRAM to a much larger dataset. For example, Choi et al (PMLH 2016; KDD 2017) trained GRAM and Doctor AI models on over 250,000 patients. These datasets are not available to us. We also note that we may also improve the performance of GRAM by modifying the code to do an unsupervised pre-training on the ~39,000 single-admission patients before the supervised training on the multi-admission patients. This goes beyond the scope of this study.

We presented the above results in **Section 3.4.2 Longitudinal EHR code prediction in MIMIC-III data** in the revision.

Application 2 on 28-year Quebec CHD dataset:

We also evaluated the code prediction accuracy on the longitudinal outpatient data from the Quebec CHD database. In this database, each patient had on average 28 years medical follow-up history (1983-2010). The input data have two data types namely, ICD-9 or ICD-10 code and procedure intervention code. We designed an RNN that takes as input both the raw EHR data and the multi-modal topic mixture (over the two ICD and procedure data types) inferred by MixEHR at a visit block t and predicts as outputs the first 3 digits of the diagnostic ICD code at the next visit block $t+1$ (**Figure R3c**).

Similar to the design from Doctor AI (Choi PMLH 2019), we defined a visit block as a one-month interval that contains at least one visit from the patient. For instance, suppose a patient visits multiple times in January and does not visit since then until May. We pool the data over those visits in January. We ignore the months where there is no visit by that patient (i.e., February to April). We predict diagnostic code for the next visit block $t + 1$ in the subsequent month that contains at least one visit by that patient (i.e., May in the above hypothetical example).

Here we sought to assess whether there is additional information provided by MixEHR that is not captured by the RNN in terms of the code prediction improvement. For the ease of reference, we call this approach as MixEHR+RNN to distinguish the baseline RNN that takes only the raw EHR as input (i.e., Doctor AI). For the baseline RNN, we use 100-dimensional hidden embedding (i.e., fully connected dense layer) to encode the observed EHR code in the input. For the MixEHR+RNN model, we used 50-dimensional hidden embedding to encode the observed EHR code augmented with 50-topic MixEHR for the topic mixture embedding (**Supplementary Fig. S21**). Therefore, both models have 100-dimensional latent embeddings to encode the input features. The rest of the architecture stays the same for both RNN models. Details are described in **Supplementary Information Section S5**.

For evaluation, we randomly split 80,000 patients into 80% of the patients for training and 20% of the patients for testing. For MixEHR+RNN, we first trained MixEHR on all of the visit blocks of the training patients. Then, we trained an RNN that takes the raw input (and the MixEHR-inferred 50-topic mixture) at each visit block of the training patients to predict the diagnostic code at the next visit block. For testing, we applied the trained MixEHR+RNN and baseline RNN to predict the diagnostic code in the next visit block of the test patients based on all of the previous visit block(s). For MixEHR+RNN and baseline RNN models, we recorded the prediction accuracy across all visit blocks (excluding the first visit block) for each patient in terms of AUROC and AUPRC for each diagnostic code.

We observed a significant improvement in terms of both AUPRC (Wilcoxon Signed Rank Tests p-value < 0.0408) and AUROC (Wilcoxon Signed Rank Tests p-value < 5.35e-55) (**Figure R3d**). Therefore, adding MixEHR significantly improves the EHR code prediction over the baseline RNN model. We also observed that the learned weights connected to the 50-topic mixture exhibit higher magnitude than the learned weights connected to the concatenated dense layer embedding (**Supplementary Fig. S19**). This means that the network relies heavily on the topic mixture to make accurate predictions. Because our dataset is focused on the CHD patients, we checked the prediction accuracy on ICD-9 code 428. Both models achieve 93% AUROC and 33% AUPRC for predicting 428 ICD-9 code (**Supplementary Data SD4**).

Compared to the baseline RNN, our MixEHR+RNN provided additional interpretation thanks to the topic model component. In particular, we discovered two interesting topics namely M43 and M1 that are highly related to heart failure ICD-9 code 428 (**Supplementary Fig. S20**). Specifically, M43 involves not only ICD-9 code 428.9 for heart failure but also code 518.4 for acute lung edema, code 290.9 for senile psychotic condition, code 428.0 for congenital heart failure, code 402.9 for hypertensive heart disease without HF, and code 782.3 for edema. Indeed, edema is known to be the precursor for heart failure among many patients. Interestingly, topic M1 characterizes a different set of heart-related diseases such as rheumatic aortic stenosis (code 395.0), secondary cardiomyopathy (code 425.9), and left heart failure (code 428.1).

Procedural codes 1-71, 1-9115, 1-9112 all indicate billing for more complex care either by a family doctor or specialists in patients who have impaired mobility, prolonged hospitalization beyond 15 days and/or admission to short-stay units for patients unable to go home, typically requiring diuretics for pulmonary edema. Therefore, most of the top EHR codes under topic M43 are clinically relevant predictors of heart-failure events in clinical practice.

Notably, our data-driven model is not protected from identifying confounder events such as 'senile psychotic condition', which does not have direct biological connection with heart-failure. We interpret this to mean that high thresholds of predictive probability should be applied to models that are purely data-driven in order to ensure optimal specificity. Additional parameters such as the inclusion of measures of time between topics that represent events could be helpful in excluding confounder events.

We presented these results in **Section 3.5** in the revision.

- For disease phenotyping/grouping tasks, performance comparisons to some of the references 29, 30, 41, 48 are necessary.

Our response: We separate our experiments on two tasks one on mortality prediction on the MIMIC-III dataset and one on the bipolar disorder classification on the Mayo Clinic dataset.

Application 1: Mortality risk prediction using the MIMIC-III dataset:

To predict mortality using the topic mixture on test patients, we first trained MixEHR on the 39K patients with single-admissions to learn the disease topics. We then used the trained MixEHR to infer topic mixture of each admission for the patients who had more than one admission. We took the last two admissions that are within 6 months apart, which gave us around 4000 patients. We used the topic mixture inferred for the second last admission as an input to a classifier to predict the mortality outcome (i.e., 'EXPIRED' in the Discharge location of the last admission). To realistically evaluate our model, we also filtered out 'Discharge summary' from the clinical notes (under CATEGORY of the notes) because they usually contain conclusions about patients' critical conditions. We then performed a 5-fold cross validation to evaluate the prediction accuracy.

We experimented with $K=50$ and 75 topics. As baseline models, we tested LDA on a flattened EHR matrix over all 6 data types, Principal Component Analysis (PCA) with 50 Principal Components (PCs), Sparsity-inducing Collected Non-negative Matrix Factorization (SiCNMF) on the same input data (Gunasekar et al., arXiv 2016). We ran SiCNMF with the default settings with lower rank set to 20. Lastly, we trained elastic net directly on the raw input data from the ~4000 patients second-last admission to predict their morality. We obtained the highest 31.62% AUPRC from MixEHR (50 topics) among all methods (**Figure R6a**). We also experimented with two different classifiers namely logistic regression with L2-norm (**Supplementary Fig. S28**) and random forest (**Supplementary Fig. S29**). We observed overall best performance using our MixEHR embedding with both classifiers compared to the other methods.

Notably, LDA obtained similar performance to MixEHR, whereas PCA, SiNMF, and EN performed a lot worse on this task. This suggests that the topic models (i.e., LDA and MixEHR) are generally more suitable to modeling the discrete count data in the EHR. Based on the elastic net linear coefficients, the 3 most positively predictive mortality topics are enriched for renal failure (M60 with the most positive coefficient), leukemia (M73), and dementia (M26), respectively (**Figure R6c**). Interestingly, the 3 most negatively predictive mortality topics are normal newborn (topic M52 with the most negative coefficient), aneurysm (M8), drug poisoning (M73), respectively.

We note that each MixEHR topic is represented by the top EHR code from diverse data types. In contrast, all of the predictive topics from LDA are represented by a single data type: the clinical notes (**Figure R6d**). This is because clinical notes contain most EHR features (i.e., words) among all of the data types. By normalizing the EHR features across all data types, the LDA topics are overwhelmed by the category that contains a lot more features than the other categories. Qualitatively, this greatly reduces the interpretability of the topics compared to our MixEHR.

We also ran Deep Patient (Miotto et al., Scientific Report 2016) on the MIMIC-III data and did not obtain good performance (**Figure R2**). This is perhaps due to the small sample size and the lack of carefully designed preprocessing pipeline as Miotto et al demonstrated on their own EHR data. Notably, in their original paper, DP was applied to a dataset containing 700,000 patients as opposed to only 58,903 admissions and only 46,501 distinct patients.

We presented the above results in **Section 3.6**.

Application 2: Classification of Bipolar Disorder using the Mayo Clinic dataset:

To further demonstrate the utility of our approach in discovering meaningful multi-modal topics, we applied MixEHR to a separate dataset containing 187 patients including 93 bipolar disorder cases and 94 age- and sex-matched controls from Mayo Clinic (see **Section 5.10.2** for more details). Despite the small sample size, the patients are deeply phenotyped: there are in total 7731 heterogeneous EHR features across 5 different data categories including ICD-9 codes, procedure codes, patient provided information (PPI), lab tests, and prescription codes. In total, there are 108,390 observations.

Among 5 data categories, lab tests and PPI have two separate data features unlike the other 3 binary data

categories. The lab and PPI data categories consist of (1) whether the data exist in EHR (e.g., whether a lab test was done or whether a patient answered a particular PPI question) and (2) actual data (e.g., test results for lab tests and patient's answer for each PPI question). Given the small sample size, we chose to model the first data feature (whether the data exist in EHR) to mimic the data structure used in the rest of data categories, recognizing that we may miss important information by not analyzing the actual lab test results and/or patient's answer for a particular PPI question. We speculate that the missing pattern may be related to the disease classification and disease subgroups, especially for the lab codes. However, we note that the missing pattern for the PPI data might be strongly confounded by other non-clinical characteristics such as age, gender and social determinants of health.

We used a 20-topic MixEHR to model the distinct distribution of these 6 data types. Our preliminary analysis showed that more than 20 topics produced degenerate models. First, we sought to quantitatively examine whether the patient topic mixtures provide useful information for accurate classification of bipolar disorder. To this end, we divided the 187 patients into five folds (**Fig. R4a**). We trained MixEHR on the four folds and then a logistic regression (LR) classifier that uses the patient topic mixture in the four folds to predict the bipolar disorder label. In this evaluation, we removed ICD-9 code category 296 from all patients as it codes for bipolar disorder. We then applied the trained MixEHR and LR to predict the BD labels of patients in the validation fold based on their MixEHR 20-topic mixture. As a comparison, we also applied two other unsupervised learning approaches, namely Latent Dirichlet Allocation (LDA) with 20 topics and Restricted Boltzmann Machine (RBM) with 20 hidden units. We observed superior performance of MixEHR+LR compared to LDA+LR and RBM+LR in both the area under the ROC and the area under the precision-recall curves (**Fig. R4b**).

These results are presented in **Section 3.3**.

a. precision-recall for future mortality prediction

b. Elastic-net linear coefficients

c. mortality-predictive topics from MixEHR (K=75)

d. mortality-predictive topics from LDA (K=75)

Figure R6. Mortality prediction. Each unsupervised embedding method was trained on the patients with only one admission in the MIMIC-III data. The trained model was then applied to embed the second last admission from patients with at least two admissions that are within 6 months apart. An elastic net (EN) classifier was trained to predict the mortality outcome in the last admission. This was performed in a 5-fold cross-validation setting. **a.** Precision-recall curve (PRC) were generated and the area under of PR (AUPRC) were displayed in the legend for each embedding method. "EN" represents the performance of elastic-net using the raw EHR features. **b.** Linear coefficients for the topics from MixEHR and LDA. The top 3 and bottom 3 topics are highlighted. **c.** Topic clinical features for the top 3 most positively predictive topics and 3 most negative predictive topics based on the elastic net coefficients for the 75 latent disease topics from MixEHR. **d.** Same as **c** but for the top predictive topics from LDA.

- For EHR/lab imputation tasks, there are also some related works other than MICE. Please review and compare in the experiments.

Our response: Thank you for the suggestion. In this revision, we demonstrated our approach on imputing the MIMIC-III lab results in comparison with the state-of-the-art collaborative filtering approach called conditional factored Restricted Boltzmann Machine (CF-RBM) (Salakhutdinov NeurIPS 2007). In contrast to traditional approaches such as multiple imputation by chained equation (MICE) (vanBuuren et al., 2011) that are not scalable to large-scale CF application, CF-RBM is scalable to large data set. This method trains a set of small RBMs only on the observed features (one RBM per user/ patient) and shares the input-hidden weights among users. CF-RBM achieved the state-of-art performance in collaborative filtering applications such as imputing Netflix movie ratings, outperforming the second best carefully fine-tuned singular vector decomposition (SVD) method. One drawback in CF-RBM is that it assumes that data are missing at random, which is an unrealistic assumption because of the aforementioned diagnosis-driven lab tests that often leads to non-missing-at-random data.

We used MixEHR with 50 topics in this experiment. For the observed lab tests of each patient, we used the topic distribution corresponding to lab test result (i.e., normal or abnormal) to infer the patient topic mixture. We further leveraged other non-lab EHR data (i.e., ICD, notes, prescription, treatment) to improve inference of the patient topic mixture (**Figure R7a** step 1). For a test patient, we masked each of his observed lab test results and inferred the topic mixture (**Figure R7a** step 2). We then found $k=25$ patients who have the lab test results observed and exhibit the most similar topic mixture to the test patient (**Figure R7a** step 3). The predicted lab result is then the average lab results over the 25 most similar training patients.

Conditioned Factored Restricted Boltzmann Machine (CF-RBM) was implemented by adapting the code from <https://github.com/felipecruz/CFRBM>. CF-RBM was originally designed for Netflix movie recommendation. Each movie has a 5-star rating. We modified the code to make it work with two-state lab results (i.e., normal and abnormal). We used 100 hidden units of CF-RBM to train on the randomly sampled 80% of the admissions and tested the CF-RBM on the remaining 20% of admissions. Here we did not distinguish whether the admissions came from the same patients but rather focused on imputing lab results within the same admission. We ensured that the training and test sets are the same for both CF-RBM and MixEHR. We evaluated the trained CF-RBM as follows. We iterated over the admissions in the test set, and for every lab test in that admission, we masked its test result and made a prediction based on the remaining lab tests of the same admission. This procedure was repeated for every lab test in every admission. We then recorded the predicted lab results and the true lab results for evaluation purposes. The same evaluation procedure was also used for MixEHR.

We observed that MixEHR achieves significantly higher accuracy compared to CF-RBM (**Figure R7b,c**; Wilcoxon one-sided test p -value < 0.00013 , Kolmogorov-Smirnov (KS) test p -value $< 1.15e-5$). This is attributable to two facts: (1) MixEHR accounts for NMAR by jointly modelling the distribution both the lab missing indicators and lab results; (2) MixEHR is able to leverage more information than CF-RBM: it models not only the lab data but also other administrative data and clinical notes.

We reviewed CF-RBM in **Section 2 Related Methods and our contributions**. We presented the results in **Section 3.5 Lab results imputation**. We described the implementation of CF-RBM in **Section 5.9**.

a. Evaluation design for imputing lab results

1. Training step: learning disease topics

2. Testing step: inferring test patient mixture

3. Predict target code t from K nearest training patients

b. Lab result imputation accuracy

c. CF-RBM versus MixEHR accuracy

Figure R7. Lab results imputation using MIMIC-III dataset. **a.** Workflow to impute lab results. Step 1. We modeled lab tests, lab test results and non-lab EHR data (i.e., ICD, notes, prescription, treatment) to infer the patient topic mixture. Step 2. For a test patient, we masked each of his observed lab test result t and inferred his topic mixture. Step 3. We then found $k = 25$ patients who have the lab test results t observed and exhibit the most similar topic mixture to the test patient. We then took the average of lab result values over the k patients as the prediction of the lab result value for the test patient j' . Steps 1-3 were repeated to evaluate every observed lab test in every test patient. **b.** We compared the imputation accuracy between MixEHR and CF-RBM. We generated the cumulative density function (CDF) of accuracy as well as the boxplot distributions (inset) for each method. In both cases, MixEHR significantly outperformed CF-RBM based on KS-test ($p < 1.15e-5$) and Wilcoxon Signed Rank one-sided test ($p < 0.00013$). **c.** CF-RBM versus MixEHR scatter plot in terms of imputation accuracy.

4. While the paper is well-written in grammar, it contains too many contents and not well-structured.

Section 4.1 - MIMIC data processing contains little information.

Our response: We reorganized and improved the structure of the paper. Overall, we made clear sections and subsections for each aspect of the manuscript to help the readers navigate the text. For instance, we put **Section 2 Related Methods** after the **Section 1 Introduction**. We also created **Section 5.9 Data description** and made an additional section namely **5.9.1 MIMIC-III**, for which we polished the text and made the description more clear.

REVIEWERS' COMMENTS:

Reviewer #1 (Remarks to the Author):

The manuscript has been improved significantly, by adding comprehensive analysis and comparative experiments on multiple datasets. This reviewer only has two minor comments.

1. Computational complexity is not analyzed. This reviewer is curious about the computing infrastructure requirement, and the computing time spent to run the proposed generative model on those datasets.

2. Due to recent interests in unsupervised machine learning techniques, many studies have developed and applied unsupervised methods to EHR data for clinical decision support or clinical research. Below is an incomplete list of recent publications. This reviewer would like to see more discussions on a comparison with those unsupervised approaches, particularly with [2] which is also a topic model.

[1]Zhao et al. "Detecting time-evolving phenotypic topics via tensor factorization on electronic health records: Cardiovascular disease case study." *Journal of biomedical informatics* 98 (2019): 103270.

[2]Wang et al. "Unsupervised machine learning for the discovery of latent disease clusters and patient subgroups using electronic health records." *Journal of Biomedical Informatics* 102 (2020): 103364.

[3]Wang et al. "The application of unsupervised deep learning in predictive models using electronic health records." *BMC Medical Research Methodology* 20.1 (2020): 1-9.

Reviewer #2 (Remarks to the Author):

The study proposes MixEHR, a multi-view Bayesian framework, for EHR data integration and modeling. The authors have significantly improved the paper and provided a comprehensive point-to-point letter. I just have some additional questions/suggestions for the experiments:

1. In the diagnostic code prediction experiment, GRAM outperforms the proposed models a lot in some ranges, especially for high-frequent range (80-100), as shown in Fig. 6(b). There should be more detailed discussions to explain the problem.

2. Moreover, the imputation experiment's is not clearly introduced. How many variables did the authors use? What are the variables' missing rates? How did the authors average 25 patients lab tests (Each patient has a sequence of lab test values)? It is unsuitable if the authors use the average of all the observed values for each variable across a long time range. It would be better if the authors compared the proposed models with some naive imputation strategies to characterize the proposed model, such as mean imputation (use the mean of current patient's observed values), and the last observed value of the same variable of the current patient. If a patient does not have any observed values for a given variable, one can use the global mean to impute it.

3. For the mortality prediction task, it is not clear how the authors select time windows. Only the early mortality prediction is meaningful in clinical settings. The first 24/48 hours data in ICU stays are usually used to predict in-hospital mortality.

Response to Reviewers

Reviewer #1 (Remarks to the Author):

The manuscript has been improved significantly, by adding comprehensive analysis and comparative experiments on multiple datasets. This reviewer only has two minor comments.

1. Computational complexity is not analyzed. This reviewer is curious about the computing infrastructure requirement, and the computing time spent to run the proposed generative model on those datasets.

Our Response: Thank you for the suggestion. In theory, the computational complexity of our model is the same as the standard LDA model: $O(DKJ)$. Here D is the number of documents (i.e., patients/admissions), K is the number of topics, and J is the size of the vocabulary (i.e., the total number of distinct EHR codes across all data types). In practice, because the actual EHR data are extremely sparse, only a very small fraction of the EHR codes is observed in any given patient's admission. Also, our model employs collapsed variational inference such that the topic inference operates on the counts of the distinct EHR code for each patient. Therefore, we can formulate the time complexity based on topics and the total number of distinct pairs of patient and EHR-code observed in the training data: $O(MK)$.

However, this does not apply for laboratory tests. The reason is because of the non-missing-at-randomness (NMAR) in the data. To account for NMAR, we not only model the observed lab results but also impute the missing lab results for every patient during the training. Therefore, the time complexity for D patients, K topics, and L lab tests each with V discrete values is $O(DKLV)$.

Together, we can conclude the time complexity for the model is $O(MK + DKLV)$, where M is the number of unique admission-EHR pairs for the non-lab test data, K is the number of topics, D is the number of patients, L is the number of lab tests, and V is the number of discrete lab test values per lab test.

In practice, our C++ implementation uses efficient numerical libraries such as Armadillo and Boost (source code: <https://github.com/li-lab-mcgill/mixehr>). We also leverage OpenMP to harness multi-core CPUs. This allows us to perform the inference step over multiple patients/admissions simultaneously (one CPU core per patient). Compared to a single-core CPU machine, we observed that the speed improvement on a multi-core CPU machine is almost linear to the number of cores. For example, if training EHR data takes 20 hours on a single core machine, training the same data set will take 2 hours on a 10-core machine.

When training a 75-topic model on the MIMIC-III data with 12-16 million observations over 50,000 admissions on a 2.2 GHz 20-core Xeon intel CPU server with 128 GB of RAM, MixEHR converges within 3 hours with 1000 iterations. We achieved similar runtime training on a 50-topic MixEHR model on the Quebec CHD data with 80,000 patients over 13 million out-patient visits for about 4 hours upon convergence. Additionally, our stochastic variational inference version of the MixEHR does not depend on the sample size N but rather updates the model on small mini-batches of patients, further improving the efficiency and scalability on large EHR dataset.

We added the above analysis to **Supplementary Methods** under **Time complexity and practical runtime**.

2. Due to recent interests in unsupervised machine learning techniques, many studies have developed and applied unsupervised methods to EHR data for clinical decision support or clinical research. Below is an incomplete list of recent publications. This reviewer would like to see more discussions on a comparison with those unsupervised approaches, particularly with [2] which is also a topic model.

[1] Zhao et al. "Detecting time-evolving phenotypic topics via tensor factorization on electronic health records: Cardiovascular disease case study." Journal of biomedical informatics 98 (2019): 103270.

[2] Wang et al. "Unsupervised machine learning for the discovery of latent disease clusters and patient subgroups using electronic health records." Journal of Biomedical Informatics 102 (2020): 103364.

[3] Wang et al. "The application of unsupervised deep learning in predictive models using electronic health records." BMC Medical Research Methodology 20.1 (2020): 1-9.

Our Response:

Thank you for the suggested papers. We have carefully read all of the 3 papers.

Zhao et al. (2019) used a tensor non-negative factorization approach to model longitudinal EHR data as a tensor with patients, PheCodes (aggregates of ICD-9 code), and time dimensions [1]. In contrast, our MixEHR models not only ICD-9 code but also laboratory tests, medication, clinical notes, procedural code, and DRG code. While we use RNN to model the longitudinal EHR data using the corresponding time-dependent topic mixture due to its efficiency, it is also possible to model the time dimension in a tensor multi-modal topic model in future works.

Wang et al. (2020) described an interesting topic model called Poisson Dirichlet Model (PDM) [2]. In contrast to the multinomial likelihood in LDA and in our MixEHR, PDM uses Poisson likelihood to model the diagnosis counts of each patient. To account for age and sex confounders, the authors introduced a weighting factor as the fitted response value from a Generalized Additive Model on sex and age. Authors showed that PDM improves topic semantics over LDA. However, because of the non-conjugacy of Dirichlet to Poisson, the PDM model inference needs to be carried out by a much slower Metropolis-Hasting MCMC sampling. This limits its application to a much smaller number of patients (below 1000 patients), a much smaller number of diagnosis codes (around 1000 codes), also a smaller number of topics (10-30 topics). In contrast, our variational collapsed Bayesian variation can easily scale to 50,000 patients and 50,000 EHR codes with 100 topics. It would be interesting to compare the topic semantics inferred by MixEHR using a large cohort with those from PDM using the allowable much smaller samples, despite the lack of explicit weighting factor of age and sex in our model. Nonetheless, the idea of age and sex correction in PDM is an important point to address in our future work.

Wang et al. (2020) used a stacked autoencoder to encode EHR codes along with their missing indicators. They then trained logistic regression on the features derived from the autoencoder to predict readmission [3]. As authors mentioned, their approach is inspired by Deep patient (Miotto et al., Scientific Report 2016), We have compared Deep patient with MixEHR in our manuscript in terms of mortality prediction and concluded that the deep learning method needs more data to be able to extract meaningful features and perform competitively in that application (**Supplementary Figure 23**).

Due to the words limit, we have concisely paraphrased the above description in the **Introduction** section of our current revision in the main text and placed the elaborate version into the **Supplementary Discussion** section in **Supplementary Information**.

Reviewer #2 (Remarks to the Author):

The study proposes MixEHR, a multi-view Bayesian framework, for EHR data integration and modeling. The authors have significantly improved the paper and provided a comprehensive point-to-point letter. I just have some additional questions/suggestions for the experiments:

1. In the diagnostic code prediction experiment, GRAM outperforms the proposed models a lot in some ranges, especially for high-frequent range (80-100), as shown in Fig. 6(b). There should be more detailed discussions to explain the problem.

Our Response:

The reason GRAM performs relatively well in this application is because it uses the multi-level CCS hierarchy as the knowledge graph, which also involves the CCS code that is predicted by the model. Also, the RNN network architecture and hyperparameters of GRAM were optimized for predicting the CCS codes whereas we used a fairly standard RNN without extensive hyperparameter tuning. As we know, neural networks have a large number of free parameters to fine-tune. This is especially important for small datasets to avoid overfitting the training data.

To investigate the latter point, we re-ran our experiments with an improved RNN architecture for the MixEHR+RNN compared to the last admission. In particular, we added a 128-dense layer to embed the input topic mixtures before passing it to the two 128-GRU layers followed by a dense layer connected to the 283 CCS medical codes as the output classes. We also chose a different set of network hyperparameters including L2 weight penalty (0.001) and dropout rate (0.5) based on existing literature. As a reference, the previous RNN architecture we used was one GRU layer (size 128) followed by another GRU layer (size 256), and then a dense layer that connects to 942 ICD-9 code output classes with L2 penalty set to 0.001 and the dropout rate set to 0.2. For GRAM and DoctorAI, we used the recommended settings that were used in their papers on the MIMIC-III data.

In terms of evaluating prediction accuracy of specific medical codes, we found that all of the models including GRAM do not perform well on the low-frequent ICD-9 codes or medical CCS codes (with median accuracies equal to 0 and mean accuracies below 25% accuracy range) and the accuracy

estimates are unstable for the rare medical codes, depending on the training and testing split. Therefore, we focused on evaluating the predictions of the 42 medical codes with at least 1000 admissions observed for the 7541 patients who had at least two admissions.

The above is described in **Methods** section **Longitudinal MIMIC-III EHR code prediction details**.

We generated a summary barplot as the average accuracy for each method (**Figure R1** or **Fig. 6b**). We observe that MixEHR+RNN confers the highest overall accuracy among the three methods although the difference between MixEHR+RNN and GRAM is small. We also generated the prediction accuracy on the codes binned by their frequencies as before (**Figure R2** or **Supplementary Figure 35**). MixEHR+RNN performs the best in 4 out of the 5 ranges and falls short only by 2% in the last range (80-100) compared to GRAM. Both MixEHR+RNN and GRAM outperform DoctorAI by a large margin.

We added the above description in the main text under the Results section **EHR code prediction in MIMIC-III data**.

2. Moreover, the imputation experiment is not clearly introduced. How many variables did the authors use? What are the variables' missing rates? How did the authors average 25 patients lab tests (Each patient has a sequence of lab test values)? It is unsuitable if the authors use the average of all the observed values for each variable across a long time range. It would be better if the authors compared the proposed models with some naive imputation strategies to characterize the proposed model, such as mean imputation (use the mean of current patient's observed values), and the last observed value of the same variable of the current patient. If a patient does not have any observed values for a given variable, one can use the global mean to impute it.

Our Response: Thank you for the suggestions. There are several points raised in this comment. Please see our point-by-point response to each of these points.

How many variables did the authors use?

Our response: In total, there are 53,432 unique features/variables over 6 data types in the MIMIC-III. Among these, 564 variables are lab tests. We used all of the variables (notes, ICD, lab, prescriptions, DRG, CPT) across all of the 6 data types to learn the topic distribution from the training set and performed imputation of lab results on the testing set.

We added this to **Methods** section **MIMIC-III lab results imputation details**

What are the variables' missing rates?

Our response: We added histograms that illustrate the frequency of observed EHR code over all admissions in the MIMIC-III data (**Figure R3** or **Supplementary Figure 27**). Majority of the variables including the lab tests were observed in less than 1% of the 58,903 admissions (i.e., greater than 99% missing rates for most variables). Therefore, the data are extremely sparse underscoring the importance of integrating multi-modal data information to aid the imputation.

We added this to **Results** section **Multimodal disease topics from MIMIC-III**

Figure R3. Frequency of EHR code in the processed MIMIC-III dataset. The histogram displays the frequency of observed EHR code over all of the admissions. Frequency of EHR code in the processed MIMIC-III dataset. The histogram displays the frequency of observed EHR code over all of the admissions.

How did the authors average 25 patients lab tests (Each patient has a sequence of lab test values)?

Our response:

To answer this question, we will first need to explain how we processed the lab data in MIMIC-III and then how we performed the imputation.

For lab test data, we used the FLAG column that indicates normal or abnormal level for the lab results. We counted for each admission the number of times the lab results were normal and abnormal. A patient at the same admission can exhibit both normal and abnormal states zero or more times for the same lab test. In this case, we recorded the frequency at each state for the same admission. This is now described in **Methods** section **MIMIC-III data description**.

Therefore, in our lab imputation analysis, we did not model the sequential event of lab test results within the same admission, which as important as it is will require a more sophisticated model that takes into account NMAR and the irregularly measured lab tests during the patient's stay. We will explore this as future work. We added this point to the **Discussion** section.

The goal of our analysis is to impute missing lab test results based on all of the other information of the patient's admission. Please note that we did not distinguish whether the admissions came from the same patients and focused on imputing lab results within the same admission. We realized that it is confusing to use the word "patient" in this analysis as we essentially operate at the *admission-level* in this experiment. We replaced the word "patient" with "admission" in the lab imputation section. Most patients only have one admission in the MIMIC-III dataset: among the 46,522 patients only 7,541 patients (16%) have more than one admission. Therefore, the results will remain the same if we were to aggregate information from multiple admissions for the same patient.

Here we randomly split the total 58,903 admissions into 80% for training and 20% for testing. For the training set, we trained our MixEHR to learn the topic distribution and the 50-topic mixture memberships for each training admission.

We then iterated over the admissions in the test set. For every lab test observed in the test admission, we first masked its test result and imputed the test result as follows:

1. We first inferred the 50-topic mixture memberships for each test admission using the topic distribution learned from the training set.
2. We then found the 25 most similar admissions to the test admission from the training set based on the Euclidean distance of their topic mixture memberships. These 25 admissions must have the target lab test observed.
3. We then took the average of the frequency over the 25 most similar training admissions as our imputed lab result for the target lab test in the test admission.

This procedure was repeated for every lab test in every test admission. We recorded the predicted lab results and the true lab results for evaluation purposes. The same way was used to evaluate the CF-RBM method.

To improve the clarity of the description, we have incorporated the above description in the revised **Methods** section under **MIMIC-III lab results imputation details**.

It is unsuitable if the authors use the average of all the observed values for each variable across a long time range.

Our response: We did not take the average of all observed values for the same patient across a long time range. Rather, we used the total counts of the observed values for each lab test within the same admission. Because MIMIC-III data contain mostly short-time admissions, most lab tests only have one observed lab test value and even when they were measured multiple times, most lab tests tend to have the same values within the same admission.

We did not model the sequential event of lab test results within the same admission, which requires a more sophisticated model that takes into account NMAR and the irregularly measured lab tests during the patient's stay. We leave these as future extensions. We added this to the **Discussion** section.

It would be better if the authors compared the proposed models with some naive imputation strategies to characterize the proposed model, such as mean imputation (use the mean of current patient's observed values), and the last observed value of the same variable of the current patient. If a patient does not have any observed values for a given variable, one can use the global mean to impute it.

Our Response: Thank you for this suggestion. To further compare our proposed approach, we took a simple imputation strategy by averaging. For each lab test, we took the average of the observed lab results over the training admissions. We then used the average frequency as the predictions for the lab results on the 20% testing admissions. We observed significantly higher accuracy for our MixEHR imputation strategy compared to the simple averaging approach (**Figure R4** or **Supplementary Figure S34**). We added this to the **Results** section **Lab results imputation using MIMIC-III data**.

between MixEHR and using the average lab values over the training admissions. We generated the cumulative density function (CDF) of accuracy as well as the boxplot distributions (inset) for each method. In both cases, MixEHR significantly outperformed CF-RBM based on KS-test ($p < 0.000176$) and Wilcoxon Signed Rank one-sided test ($p < 0.00173$). **b.** Average versus MixEHR imputation accuracy scatterplot in terms of imputation accuracy.

3. For the mortality prediction task, it is not clear how the authors select time windows. Only the early mortality prediction is meaningful in clinical settings. The first 24/48 hours data in ICU stays are usually used to predict in-hospital mortality.

Our Response:

Thank you for the suggestion. In our last revision, we took the last two admissions of each patient who has more than one admission. We limited the last two admissions within 6 months apart, which gave us around 4000 multi-admission patients. Our mortality prediction was to predict the mortality outcome in the last admission based on the patient's second last admission. We emphasized this in the **Results** section **Mortality risk prediction using the MIMIC-III data** and in the **Methods** section **MIMIC-III mortality prediction details**.

We understand the value of using 24/48 hours data to predict in-hospital mortality **within the same admission** (e.g., [17]). This is different from our application. We did not carry out this experiment because not all of the EHR data in MIMIC-III have a timestamp (i.e., CHARTTIME) within the same admission. In particular, although clinical notes and lab tests have chart time that records the time they were taken during the in-hospital admission, all of the rest of the data including ICD9 diagnostic code, prescription, CPT, DRG code do not have a chart time and are usually entered upon patient's discharge. This makes it difficult to design such an experiment that uses all of the information. We acknowledge that one caveat in our analysis is that some patients who are terminally ill (e.g., metastasis) may choose to die at home and therefore our prediction may reflect not only the patients' health states but also their mental and social states. We added this in the **Discussion section**.

We designed another experiment that used the **earliest/first admission** to predict the mortality in the **last admission**. We removed patients if their earliest admission lasted longer than 48 hours based on the difference between the admission time and discharge time. The value of this application is to identify patients who are discharged too early and therefore to provide a measure on whether the patients should be discharged from the ICU based on their current condition. This gave us around 4040 patients whose first in-hospital stay was shorter than 2 days. Same as the above, we performed a 5-fold CV on these 4040 patients as before using the pre-trained unsupervised models on the single-admission patients' data to generate latent features as input to a supervised classifier (i.e., elastic net). We observed relatively high improvement of using MixEHR compared to other baseline methods (**Figure R5** or **Supplementary Figure S33**).

We added the above to the **Results** section **Mortality risk prediction using the MIMIC-III data** and **Methods** section **MIMIC-III mortality prediction details**.

[17] Shalmali Joshi, Suriya Gunasekar, David Sontag, and Joydeep Ghosh. Identifiable Phenotyping using Constrained Non-Negative Matrix Factorization. arXiv.org, August 2016.

Figure R5. Mortality prediction in the MIMIC-III data. Each unsupervised embedding method was trained on the patients with only one admission in the MIMIC-III data. The trained model was then applied to embed the **first admission** from patients that lasted shorter than 2 days. An elastic-net (EN) classifier was trained to predict the mortality outcome **in the last admission**. This was performed in a 5-fold cross-validation setting. ROC curves were generated and the area under of the ROC (AUROC) were displayed in the legend for each embedding method. "EN" represents the performance using the raw EHR features.